# Amortized Planning with Large-Scale Transformers: A Case Study on Chess

**Anian Ruoss**[*1]     **Grégoire Delétang**[*1]     **Sourabh Medapati**[1]     **Jordi Grau-Moya**[1]

**Li Kevin Wenliang**[1]     **Elliot Catt**[1]     **John Reid**[1]     **Cannada A. Lewis**[2]     **Joel Veness**[1]

**Tim Genewein**[1]

## Abstract

This paper uses chess, a landmark planning problem in AI, to assess transformers' performance on a planning task where memorization is futile — even at a large scale. To this end, we release ChessBench, a large-scale benchmark dataset of 10 million chess games with legal move and value annotations (15 billion data points) provided by Stockfish 16, the state-of-the-art chess engine. We train transformers with up to 270 million parameters on ChessBench via supervised learning and perform extensive ablations to assess the impact of dataset size, model size, architecture type, and different prediction targets (state-values, action-values, and behavioral cloning). Our largest models learn to predict action-values for novel boards quite accurately, implying highly non-trivial generalization. Despite performing no explicit search, our resulting chess policy solves challenging chess puzzles and achieves a surprisingly strong Lichess blitz Elo of 2895 against humans (grandmaster level). We also compare to Leela Chess Zero and AlphaZero (trained without supervision via self-play) with and without search. We show that, although a remarkably good approximation of Stockfish's search-based algorithm can be distilled into large-scale transformers via supervised learning, perfect distillation is still beyond reach, thus making ChessBench well-suited for future research.

## 1 Introduction

The ability to plan ahead and reason about long-term consequences of actions is a hallmark of rationality and human intelligence. Its replication in artificial systems has been a central goal of AI research since the field's inception. Perhaps the historically most famous planning problem in AI is chess, where naive search is computationally intractable and brute-force memorization is futile — even at scale. Conceptually, the ability to plan and reason about consequences of actions is implemented via search algorithms and value computation [1, 2]. To implement such algorithms at scale, feed-forward neural networks have been playing an increasingly important role. For instance, DQN [3] and AlphaGo [4] popularized the use of neural value estimators to scale RL and (Monte Carlo) tree search. Similarly, today's strongest chess engines also employ a combination of search and amortized neural value estimation. For example, the state-of-the-art Leela Chess Zero [5, 6], which builds upon AlphaZero [7], augments Monte Carlo tree search with neural value predictions. Likewise, Stockfish 16, currently the strongest chess engine, uses efficiently updatable neural network evaluation [8], a highly specialized neural architecture to obtain fast evaluations. It is trained on value estimates of an earlier Stockfish version that uses human chess heuristics. All these chess engines tweak value-estimation networks towards fast evaluations that are to be combined with search

---

[*]Equal contribution. [1]Google DeepMind. [2]Google. Correspondence to {anianr, timgen}@google.com.

38th Conference on Neural Information Processing Systems (NeurIPS 2024).

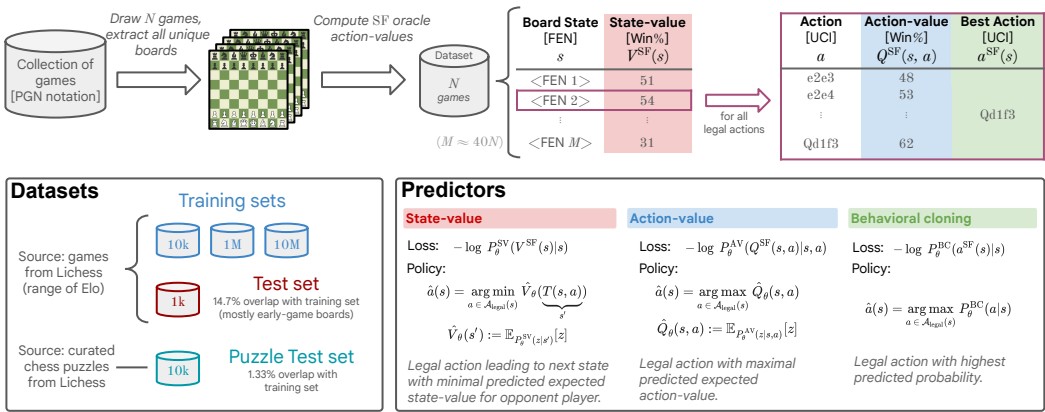

Figure 1: **Top** (Data annotation): We extract all boards from $N$ randomly drawn games from Lichess, discard duplicate board states, and compute the state-value for each board as the win-probability via Stockfish. We compute action-values and the best action for all legal moves of a board state in the same way. **Bottom left** (Dataset creation): We construct training and test sets of various sizes (see Table A1). Our largest training set has $15.3B$ action-values. Drawing games i.i.d. from the game database for our test set leads to $14.7\%$ of test boards appearing in the largest training set (mostly very early game states). We also use a test set of 10K chess puzzles that come with a correct sequence of moves. **Bottom right** (Policies): We train predictors on three targets (state- or action-values, or oracle actions), each of which can be used for a chess policy. Our value predictors are discrete discriminators (classifiers) that predict into which bin $z_i \in \{z_1, \ldots, z_K\}$ the oracle value falls.

over many possible future move sequences. As a result, the value estimates themselves may not necessarily be optimal, which leaves open the question of how far amortized planning with modern neural networks can be pushed with scale and whether the resulting "searchless" engines can match engines that use search at test time, at least in principle (putting aside computational efficiency).

To scientifically address this question, we create *ChessBench* (https://github.com/google-deepmind/searchless_chess), a large-scale chess dataset created from 10 million human games that we annotate with Stockfish 16. We use ChessBench to train transformers of up to 270 million parameters via supervised learning to predict action-values given a board-state. We also construct searchless chess policies from these predictors, where the playing strength depends entirely on the quality of the value predictions. We find that predictions by our largest trained models generalize well and non-trivially to novel board states. The resulting policies are capable of solving challenging chess puzzles and playing chess at a high level (grandmaster) against humans. Due to the combinatorial explosion of chess board states, virtually every new game involves many board states that were never seen during training (see Figure 1). This rules out memorization as a possible explanation and suggests that an approximate version of Stockfish's (search-based) value estimation algorithm can indeed be distilled into large transformers. Nonetheless, we also find that the performance gap cannot be fully closed, which may indicate that architectural innovations (or better optimization, data augmentation, etc.) might be needed instead of even larger scale.

**Our main contributions are:**

- We introduce ChessBench, a large-scale benchmark dataset for chess, consisting of $530M$ board states (from 10M games on lichess.org) annotated via Stockfish 16 with state-values and best-action, as well as $15B$ action-values for all legal actions in each board state (corresponding to roughly 8864 days of unparallelized Stockfish evaluation time).

- We train transformers of up to 270M parameters to predict action-values via supervised learning. The largest models generalize well to novel boards, enable play at grandmaster level (Lichess blitz Elo of 2895) against humans, and solve chess puzzles with difficulty ratings up to a Lichess Puzzle Elo of 2867 – without using explicit search at test time.

- We perform extensive ablations, including model- and dataset size, network architecture, action-value vs. state-value vs. behavioral cloning, and various hyper-parameters.

- We open source our ChessBench dataset, our model weights, and all training and evaluation code at `https://github.com/google-deepmind/searchless_chess`, and provide a series of benchmark results through our models, ablations, and comparisons with Stockfish 16, Leela Chess Zero, and AlphaZero, and their searchless policy/value networks.

## 2 Methodology

We now describe the dataset creation, the neural predictors and how to construct policies from them, and our evaluation methodology (see Figure 1 for an overview; full details in Appendix A).

### 2.1 ChessBench Dataset

To construct a training dataset for supervised learning we downloaded 10 million games from Lichess (lichess.org) from February 2023[1]. We extract all board states $s$ from these games and estimate the state-value $V^{\text{SF}}(s)$ for each state with Stockfish 16 (our "value oracle") using a time limit of 50ms per board (unbounded depth and maximum skill level). The value of a state is the win percentage estimated by Stockfish, lying between $0\%$ and $100\%$.[2] Note that Stockfish does not provide a centipawn score when it detects a mate-in-$k$, so we map all of these cases to a win percentage of $100\%$. We also use Stockfish to estimate action-values $Q^{\text{SF}}(s, a)$ for all legal actions $a \in \mathcal{A}_{\text{legal}}(s)$ in each state. Here we use a time limit of 50ms per state-action pair (unbounded depth and max skill level), which corresponds to a Lichess blitz Elo of 2713 for our action-value oracle (see Section 3.1). The action-values (win percentages) also determine the oracle best action $a^{\text{SF}}$:

$$a^{\text{SF}}(s) = \underset{a \in \mathcal{A}_{\text{legal}}(s)}{\arg\max} Q^{\text{SF}}(s, a).$$

In the case where multiple moves are tied in value, we pick a maximizing action arbitrarily. Since we train on individual boards and not whole games we shuffle the dataset after annotation. For our largest training dataset, based on 10M games, we thus obtain 15.3B action-value estimates (or $\approx 530\text{M}$ state-value estimates and oracle best-actions; cf. Table A1) to train on (corresponding to roughly 8864 days of unparallelized Stockfish 16 evaluation time given the limit of 50ms per move).

To create our test dataset we follow the same annotation procedure, but on 1K games downloaded from a different month (March 2023), resulting in $\approx 1.8\text{M}$ action-value estimates (or $\approx 62\text{K}$ state-value estimates and oracle best actions). Since there is only a small number of early-game board states and players often play popular openings, $14.7\%$ of boards in this i.i.d. test are also in the training set. We deliberately chose not remove them, to not introduce a distributional shift and skew test set metrics. We also create a puzzle test set, following Carlini [9], consisting of 10K challenging board states that have an Elo rating and a sequence of moves to solve the puzzle. Only $1.33\%$ of the puzzle boards appear in the training set (i.e., the initial board states, not the complete solution sequences).

### 2.2 Data Preprocessing and Training

**Value binning** The predictors we train are discrete discriminators (i.e., classifiers). Therefore we convert the win percentages (i.e., the ground truth state- or action-values) into discrete "classes" via binning, akin to distributional RL [10]. We divide the interval from $0\%$ to $100\%$ uniformly into $K$ bins (non-overlapping sub-intervals) and assign a one-hot code to each bin $z_i \in \{z_1, \ldots, z_K\}$. If not mentioned otherwise, $K = 128$. For our behavioral cloning experiments we train to directly predict the oracle actions, which are already discrete. We ablate the number of bins $K$ in Appendix B.1, and we investigate losses that do (i.e., the $\log$ loss) and do not (i.e., the HL-Gauss and L2 loss) maintain semantic interconnectedness of the labels (i.e., the dependence between the classes) in Table 2.

**Tokenization** A FEN [11] string is a standard string-based description of a chess position. It consists of a board state, the current side to move, the castling availability for both players, a potential

---

[1]Lichess games and puzzles are released under the Creative Commons CC0 license.

[2]Stockfish returns a score in centipawns that we convert to the win percentage with the standard formula $\text{win}\% = 100/(1 + \exp(-0.00368208 \cdot \text{centipawns}))$ from `https://lichess.org/page/accuracy`.

*en passant* target, a half-move clock and a full-move counter, all represented in a single ASCII string. Our tokenization uses this representation, except that we flatten any uses of run-length encoding to obtain a fixed length (77) tokenized representation. Actions are stored in UCI notation [12], for example 'e2e4' for the popular opening move for white. To tokenize we use the index of a move into a lexicographically sorted array of the 1968 possible UCI encoded actions (see Appendix A.1 for more details). Note that this representation technically makes the game non-Markovian, because FENs do not contain the move history and therefore, cannot capture all information for rules such as drawing by threefold repetition (drawing because the same board occurs three times).

**Network inputs and outputs**  For all our predictors we use a modern decoder-only transformer backbone [13–15] to parameterize a categorical distribution by normalizing the transformer's outputs with a $\log \mathrm{softmax}$ layer. The models thus output $\log$ probabilities. The context size is 79 for action-value prediction and 78 for state-value prediction and behavioral cloning (see 'Tokenization' above). The output size is $K$ (i.e., the number of bins) for action- and state-value prediction and 1968 (the number of all possible legal actions) for behavioral cloning. We use learned positional encodings [16] as the length of the input sequences is constant. Our largest model has $\approx 270$ million parameters, i.e., 16 layers, 8 heads, and an embedding dimension of 1024 (full details in Appendix A.2).

**Training protocol**  We train our predictors by minimizing the cross-entropy loss via mini-batch stochastic gradient descent using Adam [17]. The target labels are either bin-indices for state- or action-value prediction or action indices for behavioral cloning. For state- and action-value prediction, we additionally apply label smoothing via the HL-Gauss loss [18], using a Gaussian smoothing distribution with the mean given by the label and a standard deviation of $\sigma = 0.75/K \approx 0.05$ as recommended by Farebrother et al. [19]. Thus, the labels are no longer one-hot but multi-hot encoded (see Figure 3 in [19] for an overview). We ablate the loss function, i.e., HL-Gauss vs. cross-entropy vs. MSE, in Section 3.4. We train for 10 million steps, which corresponds to 2.67 epochs for a batch size of 4096 with 15.3B data points (cf. Table A1; details in Appendices A.2 and A.3).

## 2.3   Predictors and Policies

Our predictors are categorical distributions parameterized by neural networks $P_\theta(z|x)$ that take a tokenized input $x$ and output a predictive distribution over discrete labels $\{z_1, \ldots, z_K\}$. Depending on the prediction target we distinguish between three tasks (see Figure 1 for an overview).

**(AV) Action-value prediction**  The target label is the bin $z_i$ into which the ground-truth action-value estimate $Q^{\mathrm{SF}}(s, a)$ falls. The input to the predictor is the concatenation of tokenized state and action. The loss for a single input data point $(s_i, a_i)$ is:

$$ - \sum_{z \in \{z_1, \ldots z_K\}} q_i(z) \log P_\theta^{\mathrm{AV}}(z|s_i, a_i) \quad \text{with } q_i := \text{HL-Gauss}_K(Q^{\mathrm{SF}}(s, a)), \tag{1} $$

where $K$ is the number of bins and HL-Gauss$_K(x)$ is the function that computes $q_i$ (the smoothed version of label $z_i$), i.e., a smooth categorical distribution using HL-Gauss [18] rather than the standard one-hot procedure. To use the predictor in a policy, we evaluate the predictor for all legal actions in the current state and pick the action with maximal expected action-value:

$$ \hat{a}^{\mathrm{AV}}(s) = \underset{a \in \mathcal{A}_{\mathrm{legal}}}{\arg \max} \underbrace{\mathbb{E}_{Z \sim P_\theta^{\mathrm{AV}}(\cdot|s,a)}[Z]}_{\hat{Q}_\theta(s,a)}. $$

**(SV) State-value prediction**  The target label is the bin $z_i$ that the ground-truth state-value $V^{\mathrm{SF}}(s)$ falls into. The input to the predictor is the tokenized state. The loss for a single data point is:

$$ - \sum_{z \in \{z_1 \ldots z_K\}} q_i(z) \log P_\theta^{\mathrm{SV}}(z|s_i) \quad \text{with } q_i := \text{HL-Gauss}_K(V^{\mathrm{SF}}(s_i)). \tag{2} $$

To use the state-value predictor as a policy, we evaluate the predictor for all states $s' = T(s, a)$ that are reachable via legal actions from the current state (where $T(s, a)$ is the deterministic transition of taking action $a$ in state $s$). Since $s'$ implies that it is now the opponent's turn, the policy picks the action that leads to the state with the worst expected value for the opponent:

$$ \hat{a}^{\mathrm{SV}}(s) = \underset{a \in \mathcal{A}_{\mathrm{legal}}}{\arg \min} \underbrace{\mathbb{E}_{Z \sim P_\theta^{\mathrm{SV}}(\cdot|s')}[Z]}_{\hat{V}_\theta(s')}. $$

**(BC) Behavioral cloning** The target label is the (one-hot) action-index of the ground-truth action $a^{\text{SF}}(s)$ within the set of all possible actions (see 'Tokenization' in Section 2.2). The input to the predictor is the tokenized state, which leads to the loss for a single data point:

$$-\log P_\theta^{\text{BC}}(a^{\text{SF}}(s)|s). \tag{3}$$

This gives a policy that picks the highest-probability action:

$$\hat{a}^{\text{BC}}(s) = \arg\max_{a \in \mathcal{A}_{\text{legal}}} P_\theta^{\text{BC}}(a|s).$$

### 2.4 Evaluation

We use the following metrics to evaluate our models and/or measure training progress.

**Action accuracy** The test set percentage where the policy picks the best action: $\hat{a}(s) = a^{\text{SF}}(s)$.

**Action ranking (Kendall's $\tau$)** The average test set rank correlation [20] of the predicted action distribution with the ground truth given by Stockfish. Kendall's $\tau$ ranges from -1 (exact inverse order) to 1 (exact same order), with 0 indicating no rank correlation. The ranking is obtained by evaluating for all legal actions $\hat{Q}_\theta(s, a)$, $-\hat{V}_\theta(T(s, a))$, or $P_\theta^{\text{BC}}(a|s)$, in the case of AV, SV, or BC prediction, respectively. The ground-truth ranking is given by evaluating $Q^{\text{SF}}(s, a)$ for all legal actions.

**Puzzle accuracy** We evaluate our policies on their capability of solving puzzles from a collection of Lichess puzzles that are rated by Elo difficulty from 399 to 2867 based on how often each puzzle has been solved correctly. We use *puzzle accuracy* as the percentage of puzzles where the policy's action sequence exactly matches the *entire* solution action sequence. For our main results in Sections 3.1 and 3.2 we use 10K puzzles; otherwise, we use the first 1K puzzles to speed up evaluation.

**Game playing strength (Elo)** We evaluate the playing strength (measured as an Elo rating) of the policies in two ways: (i) we run an internal tournament between all the agents from Table 1 except for GPT-3.5-turbo-instruct, and (ii) we play Blitz games on Lichess against either only humans or only bots. For the tournament we play 400 games per agent pair, yielding 22K games in total, and compute the Elo with BayesElo [21] using the default confidence parameter of 0.5. To ensure variability in the games, we use the openings from the Encyclopaedia of Chess Openings (ECO) [22]. When playing on Lichess, we use a softmax policy with a low temperature of 0.005 for the first five full-moves instead of the arg max policy to create variety in games and prevent simple exploits via repeated play. We anchor the relative BayesElo values to the Lichess Elo vs. bots of our largest (270M) model.

### 2.5 Engine Comparisons

We compare the performance of our models against the following engines:

**Stockfish 16** We consider two variants: (i) a 50ms time limit *per legal move* (i.e., the dataset oracle), and (ii) a 1.5s limit *per board*, which is roughly the amount of time the first variant (i) takes on average per board (i.e., there are roughly 30 legal moves per board on average).

**AlphaZero** We consider three variants of AlphaZero [7]: (i) with 400 MCTS simulations, (ii) only the policy network, and (iii) only the value network (where (ii) and (iii) perform no additional search). AlphaZero's networks have 27.6M parameters and are trained on 44M games (full details in [23]).

**Leela Chess Zero** We consider three variants: (i) with 400 MCTS simulations, (ii) only the policy network, and (iii) only the value network (where (ii) and (iii) perform no additional search). We use the T82 network, which is a convolutional network with 768 filters, 15 blocks, and mish activations [24] – the largest network available on the official Leela Chess Zero website. At the time of writing, the precise network architecture, training data, and training protocol for the T82 network were not reported on the Leela Chess Zero website or other source. Concurrently to our paper, Monroe and Leela Chess Zero Team [6] published a rigorous tech report which now provides many of these details about for a slightly different (state-of-the-art) Leela Zero architecture called ChessFormer, and includes a comparison against our vanilla networks.

Table 1: Comparison of our action-value models against Stockfish 16, variants of Leela Chess Zero and AlphaZero (with and without Monte Carlo tree search), and GPT-3.5-turbo-instruct. Tournament Elo ratings are obtained by making the agents play against each other and cannot be directly compared to the Lichess Elo. Lichess (blitz) Elo ratings result from playing against either human opponents or bots on Lichess. Stockfish 16 with a time limit of 50ms per move is our data-generating oracle. Models operating on the PGN observe the full move history, whereas FENs only contain very limited historical information (sufficient for the fifty-move rule). Unlike all other engines, our policies were trained with supervised learning and use no explicit search at test time (except for GPT-3.5-turbo-instruct, which was trained via self-supervised learning and then instruction tuned).

| | | | | | Lichess Elo | | |
| Agent | Train | Search | Input | Tournament Elo | vs. Bots | vs. Humans | Puzzle Acc. (%) |
|---|---|---|---|---|---|---|---|
| 9M Transformer (ours) | SL | | FEN | 2025 ($\pm$18) | 2054 | - | 88.9 |
| 136M Transformer (ours) | SL | | FEN | 2259 ($\pm$16) | 2156 | - | 94.5 |
| 270M Transformer (ours) | SL | | FEN | 2299 ($\pm$15) | 2299 | 2895 | 95.4 |
| GPT-3.5-turbo-instruct | SSL | | PGN | - | 1755 | - | 66.5 |
| AlphaZero (policy net only) | RL | | PGN | 1777 ($\pm$25) | - | - | 56.1 |
| AlphaZero (value net only) | RL | | PGN | 1992 ($\pm$19) | - | - | 82.0 |
| AlphaZero (400 MCTS sim.) | RL | ✓ | PGN | 2470 ($\pm$16) | - | - | 95.6 |
| Leela Chess Zero (policy net only) | RL | | PGN | 2292 ($\pm$16) | 2224 | - | 88.6 |
| Leela Chess Zero (value net only) | RL | | PGN | 2418 ($\pm$16) | 2318 | - | 95.9 |
| Leela Chess Zero (400 MCTS sim.) | RL | ✓ | PGN | 2858 ($\pm$20) | 2620 | - | 99.6 |
| Stockfish 16 (50ms per move) [oracle] | SL | ✓ | FEN + Moves | 2711 ($\pm$18) | 2713 | - | 99.8 |
| Stockfish 16 (1.5s per board) | SL | ✓ | FEN + Moves | 2935 ($\pm$23) | 2940 | - | 100.0 |

**GPT-3.5-turbo-instruct** We follow Carlini [9] and encode the entire game with the Portable Game Notation (PGN) [11] to reduce hallucinations. Since Carlini [9] found that (at that time) GPT-4 struggled to play full games without making illegal moves, we do not consider GPT-4.

In contrast to our models, all of the above engines rely on the entire game's history (via the PGN; we use the FEN, which only contains the game's current state and very limited historical information). Observing the full game's history helps, for instance, detecting and preventing draws from threefold repetition (games are drawn if the same board state appears three times throughout the game). Our engines require a workaround to deal with this problem (described in Section 4).

A direct comparison between all engines comes with a lot of caveats since some engines use the game history, some have very different training protocols (i.e., RL via self-play instead of supervised learning), and some use search at test time. We show these comparisons to situate the performance of our models within the wider landscape, but emphasize that some conclusions can only be drawn within our family of models and the corresponding ablations that keep all other factors fixed.

## 3 Case-Study Results

We use two settings for our experiments: (i) a large-scale setting for our main results (Sections 3.1 and 3.2; details in Appendix A.2), and (ii) a setting geared towards getting representative results with better computational efficiency for our ablations (Sections 3.3 and 3.4; details in Appendix A.3).

### 3.1 Main Result

Table 1 shows the playing strength (internal tournament Elo, external Lichess Elo, and puzzle accuracy) of our large-scale transformers trained on the full (10M games) training set. We compare models with 9M, 136M, and 270M parameters (none of them overfit the training set as shown in Appendix B.4). The results show that all three models exhibit non-trivial generalization to novel boards and can successfully solve a large fraction of puzzles. Across all metrics, increasing model size consistently improves the scores, confirming that model scale matters for strong chess performance. Our largest model achieves a blitz Elo of 2895 against human players, which places it into grandmaster territory. However, the Elo drops when playing against bots on Lichess, which may be a result of having a significantly different player pool [25], minor technical issues, or a qualitative difference in how bots exploit weaknesses compared to humans (see Section 4 for a detailed discussion).

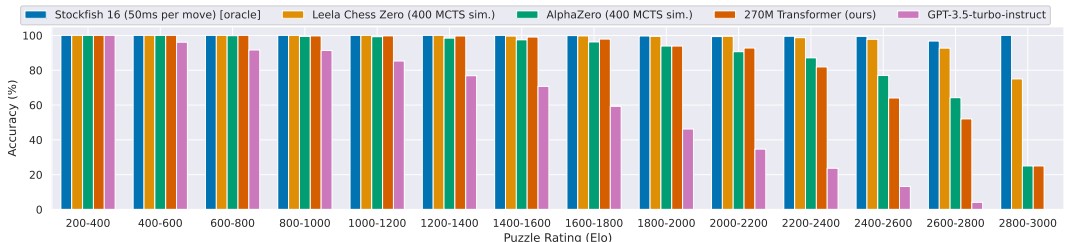

Figure 2: Puzzle solving comparison for our 270M transformer, Stockfish 16 (50ms per move), Leela Chess Zero, AlphaZero, and GPT-3.5-turbo-instruct on 10K Lichess puzzles (curated following [9]).

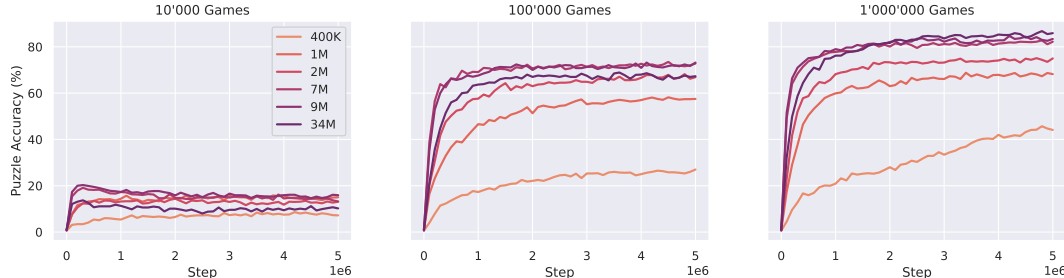

Figure 3: Puzzle accuracy for different training set sizes (stated above panels) and model sizes (color-coded), evaluated on our small set of 1K puzzles. Generally, larger models trained on larger datasets lead to higher accuracy (which strongly correlates with test set performance and general chess playing strength), highlighting the importance of scale for strong chess play. This effect cannot be explained by memorization since $< 1.41\%$ of the initial puzzle board states appear in our training set. If the model is too large in relation to the training set it overfits (left panel; loss curves in Figure A3).

## 3.2 Puzzles

In Figure 2 we compare the puzzle performance of our 270M parameter model against Stockfish 16 (50ms limit per move), GPT-3.5-turbo-instruct, AlphaZero, and Leela Chess Zero. We use our large puzzle set of 10K puzzles, grouped by their assigned Elo difficulty from Lichess. Stockfish 16 performs the best across all difficulty categories, followed by Leela Chess Zero, AlphaZero, and our model. Impressively, our model, which does not use any explicit search at test time, nearly matches the performance of AlphaZero with (Monte Carlo) tree search. GPT-3.5-turbo-instruct achieves non-trivial puzzle performance but significantly lags behind our model. We emphasize that solving the puzzles requires a correct move *sequence*, and since our policy cannot explicitly plan ahead, solving the puzzle sequences relies entirely on having good value estimates that can be used greedily.

## 3.3 Scaling Analysis

Figure 3 shows a scaling analysis over the dataset and model size. We visualize the puzzle accuracy (train and test losses in Figure A3), which correlates well with the other metrics and the overall playing strength. For small training set size (10K games, left panel) larger architectures ($\geq$ 7M) start to overfit. This effect disappears as the dataset size is increased to 100K (middle panel) and 1M games (right panel). The results also show that the final accuracy of a model increases as the dataset size is increased (consistently across model sizes). Similarly, we observe the general trend of increased architecture size leading to increased overall performance (as in our main result in Section 3.1).

## 3.4 Variants and Ablations

We perform extensive ablations using the 9M-parameter model and show the results in Tables 2 and A2. We use the results and conclusions drawn to inform and justify our design choices and determine the default model-, data-, and training configurations.

**Predictor targets** By default we learn to predict action-values given a board state. Here we compare against using state-values or oracle actions (behavioral cloning) as the prediction targets (Section 2.3 and Figure 1 describe how to construct policies from the predictors). Table 2 and Figure A4 show that the action-value predictor is superior in terms of action-ranking (Kendall's $\tau$), action accuracy, and puzzle accuracy. This superior performance might stem primarily from the significantly larger action-value dataset (15.3B state-action pairs vs. $\approx$ 530M states for our largest training set). We thus run an ablation where we train all three predictors on the same amount of data (results in Table A4 and Figure A5, which largely confirm this hypothesis). Appendix B.5 contains detailed discussion, including why the performance discrepancy between behavioral cloning and the state-value prediction policy may be largely due to training only on expert actions rather than the full action distribution.

Table 2: Ablating the predictor target, loss function, network depth, and number of value bins (see Section 3.4). The best configurations are: action-value prediction (see Appendix B.5 for a detailed discussion), HL-Gauss loss, depth 16, and 128 bins. We conduct further ablations (over the data sampler, the Stockfish time limit, and the model architecture) in Appendix B.1 and Table A2.

| | | Accuracy (%) | | |
|---|---|---|---|---|
| **Ablation** | **Parameter** | **Puzzles** | **Actions** | **Kendall's $\tau$** |
| Predictor target | Action-Value | **83.3** | **63.0** | **0.259** |
| | State-Value | 77.5 | 58.5 | 0.215 |
| | Behavioral Cloning | 65.7 | 56.7 | 0.116 |
| Loss function | HL-Gauss (class.) | **82.0** | 61.8 | **0.257** |
| | log (class.) | 80.6 | **61.9** | **0.257** |
| | L2 (regr.) | 80.8 | 58.9 | 0.240 |
| Network depth | 2 | 54.7 | 51.5 | 0.209 |
| | 4 | 40.5 | 44.3 | 0.179 |
| | 8 | 79.5 | 60.7 | 0.252 |
| | 16 | **81.3** | **61.6** | **0.256** |
| | 32 | 79.5 | 61.2 | 0.254 |
| No. of value bins | 16 | 83.0 | 61.4 | 0.248 |
| | 32 | 83.0 | 63.2 | 0.261 |
| | 64 | **84.4** | 63.1 | 0.259 |
| | 128 | 83.8 | **63.4** | **0.262** |
| | 256 | 83.7 | 63.0 | 0.260 |

**Loss function** We treat learning Stockfish action-values as a classification problem and train by minimizing the HL-Gauss loss [18]. Here we compare this to the cross-entropy loss (log-loss), which is as close as possible to the (tried and tested) standard LLM setup. Another alternative is to treat the problem as a scalar regression problem by parameterizing a fixed-variance Gaussian likelihood model with a transformer and performing maximum (log) likelihood estimation, i.e., minimizing the mean-squared error (L2 loss). To that end, we modify the architecture to output a scalar (without a final log-layer or similar). Table 2 shows that the HL-Gauss loss outperforms the other two losses.

**Network depth** Table 2 shows the influence of increasing the transformer's depth (i.e., the number of layers) for a fixed number of parameters (we vary the embedding dimension and the widening factor such that all models have the same number of parameters). Since transformers may learn to roll out iterative computation (which arises in search) across layers, deeper networks may hold the potential for deeper unrolls. The performance of our models increases with their depth but saturates at around 16 layers, indicating that depth is important, but not beyond a certain point.

**Value binning** Table 2 shows the impact of varying the number of bins used for state- and action-value discretization (from 16 to 256), demonstrating that more bins generally lead to improved performance (up to a certain point). We therefore use $K = 128$ bins for our experiments.

# 4 Discussion

When using our state-based policies to play against humans and bots, two minor technical issues appear that can only be solved by having (some) access to the game's history. In this section, we discuss both issues and present our corresponding workarounds.

**Blindness to threefold repetition** By construction, our state-based predictor cannot detect the risk of threefold repetition (drawing because the same board occurs three times), since it has no access to the game's history (FENs contain minimal historical info, sufficient for the fifty-move rule). To reduce draws from threefold repetitions, we check if the agent's next move would trigger the rule and set the corresponding action's win percentage to $50\%$ before computing the softmax. However, our agents still cannot plan ahead to mimimize the risk of being forced into threefold repetition.

**Indecisiveness in the face of overwhelming victory** If Stockfish detects a mate-in-$k$ it outputs $k$ and not a centipawn score. When annotating our dataset, we map all such outputs to a win percentage of $100\%$. Similarly, in a very strong position, several actions may end up in the maximum value bin. Thus our agent sometimes plays somewhat randomly rather than committing to a plan that finishes the game (the agent has no knowledge of its past moves). Paradoxically, this means that our agent, despite being in a position of overwhelming win percentage, sometimes fails to take the (virtually) guaranteed win (see Figure 4) and might draw or even end up losing since small chances of a mistake accumulate with longer games. To alleviate this, we check whether the predicted scores for all top five moves lie above a win percentage of $99\%$ and double-check this condition with Stockfish, and if so, use Stockfish's top move (out of these) to have consistency in strategy across time-steps.

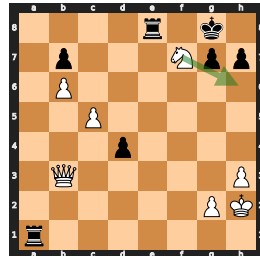 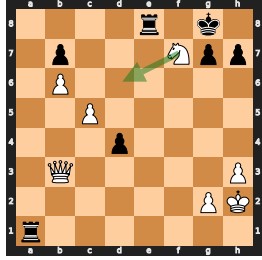

(a) Possible Move (Mate-in-3)  (b) Actual Move (Mate-in-5)

Figure 4: Two options to win the game in 3 or 5 moves, respectively (more options exist). Since they both map into the highest-value bin our bot ignores Nh6+, the fastest way to win (in 3), and instead plays Nd6+ (mate-in-5). Unfortunately, a state-based predictor without explicit search cannot guarantee that it will continue playing the Nd6+ strategy and thus might randomly alternate between different strategies. Overall this increases the risk of drawing the game or losing due to a subsequent (low-probability) mistake, such as a bad $\mathrm{softmax}$ sample. Board from a game between our 9M Transformer (white) and a human (blitz Elo of $2145$).

**Elo: Humans vs. bots** We have three plausible hypotheses for why the Lichess Elo in Table 1 differs when playing against humans vs. bots: (i) humans tend to resign when our bot is in an overwhelmingly winning position but bots do not (i.e., the previously described problem gets amplified against bots); (ii) most humans on Lichess rarely play against bots, i.e., the two player pools (humans and bots) are hard to compare and their Elo ratings may be miscalibrated [25]; and (iii) based on anecdotal analysis by a chess National Master, our models make the occasional tactical mistake which may be penalized more severely by bots than humans (analysis in Appendices B.7 and B.8). While investigating this Elo discrepancy is interesting, it is not central to our paper and does not impact our main claims.

## 4.1 Limitations

Our primary goal was to investigate whether a complex search algorithm such as Stockfish 16 can be approximated with a feedforward neural network on our dataset via supervised learning. While our largest model achieves good performance, it does not fully close the gap to Stockfish 16, and it is unclear whether further scaling would close this gap or whether other innovations are needed.

While we produced a strong chess policy, our goal was not to build a state-of-the-art chess engine. Our models are impractical in terms of speed and would perform poorly in computer chess tournaments with compute limitations. Therefore, we calibrated our policy's playing strength via Lichess, where the claim of "grandmaster-level" play currently holds only against human opponents. In addition, we only evaluated our biggest model against humans on Lichess due to the extensive amount of time required. We also cannot rule out that opponents, through extensive repeated play, may find weaknesses due to the deterministic nature of our policy. Finally, we reiterate that we compare to other engines to situate our models within the wider landscape, but that a direct comparison between all the engines comes with a lot of caveats due to differences in their inputs (FENs vs. PGNs), training protocols (RL vs. supervised learning), and the use of search at test time.

Leela Chess Zero's networks, which are trained with self-play and RL, achieve higher Elo ratings without using explicit search at test time than our transformers, which we trained via supervised learning. However, in contrast to our work, very strong chess performance (at low computational cost) is the explicit goal of this open source project (which they have clearly achieved via domain-specific adaptations). We refer interested readers to [6] (which was published concurrently to our work) for details on the current state-of-the-art and a comparison against our networks.

## 5  Related Work

Unsurprisingly given its long history in AI, there is a vast literature on applying algorithmic techniques to chess. Earlier works predominantly focused on methods which would increase the strength of chess playing entities, but as time has progressed, the role of chess as a problem domain has changed more from a challenge area to that of an illuminating benchmark, and our work continues this tradition.

Early computer chess research focused on designing explicit search strategies coupled with heuristics, as evidenced by Turing's initial explorations [26] and implementations like NeuroChess [27]. This culminated in systems like Deep Blue [28] and early versions of Stockfish [29].

AlphaZero [7] and Leela Chess Zero [5] marked a paradigm shift: They employed deep RL with Monte Carlo tree search to learn heuristics (policy and value networks) instead of manually designing them [30, 31]. Several works built upon this framework [32], including enhancements to AlphaZero's self-play mechanisms [33] and the use of model-free RL [23]. At the time of writing, Leela Zero's T82 networks were state-of-the art, but their training protocol, training data, and some architectural details were not fully reported. In the meantime, Monroe and Leela Chess Zero Team [6] have published a detailed tech report on another transformer-based Leela Zero architecture called the ChessFormer. Their comparison shows that ChessFormers comparable in size to our models outperform our vanilla transformers while requiring fewer FLOPS thanks to clever domain-specific adaptations.

Another line of work moved away from explicit search methods by leveraging large-scale game datasets for (un)supervised learning, both for chess [34–37] and Go [38]. Most closely related to our work, project Maia [35] used behavioral cloning on 12M human games but, rather than maximizing performance (i.e., distilling Stockfish), focused on predicting human moves at different levels of play.

The rise of large (pretrained) foundation models also led to innovations in various areas of computer chess research: learning the rules of chess [39, 40], evaluating move quality [41], evaluating playing strength [9, 42], state tracking [43, 44], and playing chess from visual inputs [45]. Fine-tuning on chess-specific data sources (e.g., chess textbooks) has further improved performance [46–48].

## 6  Conclusion

Our paper introduces ChessBench, a large-scale, open source benchmark dataset for chess, and shows the feasibility of distilling an approximation of Stockfish 16, a complex planning algorithm, into a feed-forward transformer via standard supervised training. The resulting predictor generalizes well to unseen board states, and, when used in a policy, leads to strong chess play. We show that strong chess capabilities from supervised learning only emerge at sufficient dataset and model scale. Our work thus adds to a rapidly growing body of literature showing that sophisticated algorithms can be distilled into feed-forward transformers, implying a paradigm shift to viewing large transformers as a powerful technique for general algorithm approximation rather than "mere" statistical pattern recognizers. Nevertheless, perfect distillation of Stockfish 16 is still beyond reach and closing the performance gap might need other (e.g., architectural) innovations. Our open source benchmark dataset, ChessBench, thus provides a solid basis for scientific comparison of any such developments.

## 7  Impact Statement

While the results of training transformer-based architectures at scale in a (self-)supervised way will have significant societal consequences in the near future, these concerns do not apply to closed domains, such as chess, that have limited real-world impact. Moreover, chess has been a domain of machine superiority for decades. Another advantage of supervised training on a single task over other forms of training (particularly self-play or reinforcement learning and meta-learning) is that the method requires a strong oracle solution to begin with (for data annotation) and is unlikely to significantly outperform the oracle, which means that the potential for the method to rapidly introduce substantial unknown capabilities (with wide societal impacts) is very limited.

## Acknowledgments

We thank Aurélien Pomini, Avraham Ruderman, Eric Malmi, Charlie Beattie, Chris Colen, Chris Wolff, David Budden, Dashiell Shaw, Guillaume Desjardins, Guy Leroy, Hamdanil Rasyid, Himanshu Raj, John Schultz, Julian Schrittwieser, Laurent Orseau, Lisa Schut, Marc Lanctot, Marcus Hutter, Matthew Aitchison, Matthew Sadler, Nando de Freitas, Neel Nanda, Nenad Tomasev, Nicholas Carlini, Nick Birnie, Nikolas De Giorgis, Ritvars Reimanis, Satinder Baveja, Thomas Fulmer, Tor Lattimore, Vincent Tjeng, Vivek Veeriah, Zhengdong Wang, and the anonymous reviewers for insightful discussions and their helpful feedback.

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

# A Experimental Setup

## A.1 Tokenization

The first part of a FEN string encodes the position of pieces rank-wise (row-wise). The only change we make is that we encode each empty square with a '.', which always gives us 64 characters for a board. The next character denotes the active player ('w' or 'b'). The next part of the FEN string denotes castling availability (up to four characters for King- and Queen-side for each color, or '-' for no availability)—we take this string and if needed pad it with '.' such that it always has length 4. Next are two characters for the *en passant* target, which can be '-' for no target; we use the two characters literally or '-.' for no target. Finally we have the halfmove clock (up to two digits) and the fullmove number (up to three digits); we take the numbers as characters and pad them with '.' to make sure they are always tokenized into two and three characters respectively.

## A.2 Main Setup

We use the same basic setup for all our main experiments and only vary the model architecture.

We train for 10 million steps with a batch size of 4096, meaning that we train for 2.67 epochs on the full training dataset (10M games). We use the Adam optimizer [17] with a learning rate of $1 \times 10^{-4}$. We train on the dataset generated from 10 million games (cf. Table A1) for the action-value policy with 128 return buckets and a stockfish time limit of 0.05s. We use the unique sampler and evaluate on 1k games (cf. Table A1) and 10k puzzles from a different month than that used for training.

We train a vanilla decoder-only transformer without causal masking [13], with the improvements proposed in LLaMA [14] and Llama 2 [15], i.e., post-normalization and SwiGLU [49]. We use three different model configurations (with a widening factor of 4): (i) 8 heads, 8 layers, and an embedding dimension of 256, (ii) 8 heads, 8 layers, and an embedding dimension of 1024, and (iii) 8 heads, 16 layers, and an embedding dimension of 1024.

## A.3 Ablation Setup

We use the same basic setup for all our ablation experiments and only vary the ablation parameters.

We train for 5 million steps with a batch size of 1024, meaning that we train for 3.19 epochs on a smaller training dataset (1M games). We use the Adam optimizer [17] with a learning rate of $4 \times 10^{-4}$. We train on the dataset generated from 1 million games (cf. Table A1) for the action-value policy with 32 return buckets and a stockfish time limit of 50ms. We use the unique sampler and evaluate on 1K games (cf. Table A1) and 1K puzzles (1.4% overlap with training set) from a different month than that used for training. We train a vanilla decoder-only transformer [13] with post-normalization, 8 heads, 8 layers, an embedding dimension of 256, a widening factor of 4, and no causal masking.

## A.4 Dataset Statistics

We visualize some dataset statistics in Figure A1. Note that, even though we should have the same amount of board-state data points for state-value prediction and behavioral cloning, we occasionally run into time-outs when annotating our dataset with Stockfish (due to our internal infrastructure setup) and therefore drop the corresponding record.

## A.5 Playing-Strength Evaluation

**Lichess** We evaluate and calibrate the playing strength of our models by playing against humans and bots on Lichess (using the lichess-bot project [50] as a basis). Our standard evaluation allows for both playing against bots and humans (see Table 1), but since humans tend to rarely play against bots the Elo ratings in this case are dominated by playing against other bots (see our discussion of how this essentially creates two different, somewhat miscalibrated, player pools in Section 4). In our case the policies in the column denoted with 'vs. Bots' in Table 1 have played against some humans but the number of games against humans is $< 4.5\%$ of total games played. To get better calibration against humans we let our largest model play exclusively against humans (by not accepting games with other bots) which leads to a significantly higher Elo ranking (see Table 1). Overall we have

Table A1: Dataset sizes. For simplicity, we typically refer to the datasets by the number of games they were created from.

| Split | Games | State-Value | | Behavioral Cloning | | Action-Value | |
|---|---|---|---|---|---|---|---|
| | | Records | Bytes | Records | Bytes | Records | Bytes |
| Train | $10^4$ | 591 897 | 43.7 MB | 589 130 | 41.1 MB | 17 373 887 | 1.4 GB |
| | $10^5$ | 5 747 753 | 422.0 MB | 5 720 672 | 397.4 MB | 167 912 926 | 13.5 GB |
| | $10^6$ | 55 259 971 | 4.0 GB | 54 991 050 | 3.8 GB | 1 606 372 407 | 129.0 GB |
| | $10^7$ | 530 310 443 | 38.6 GB | 527 633 465 | 36.3 GB | 15 316 914 724 | 1.2 TB |
| Test | $10^3$ | 62 829 | 4.6 MB | 62 561 | 4.4 MB | 1 838 218 | 148.3 MB |

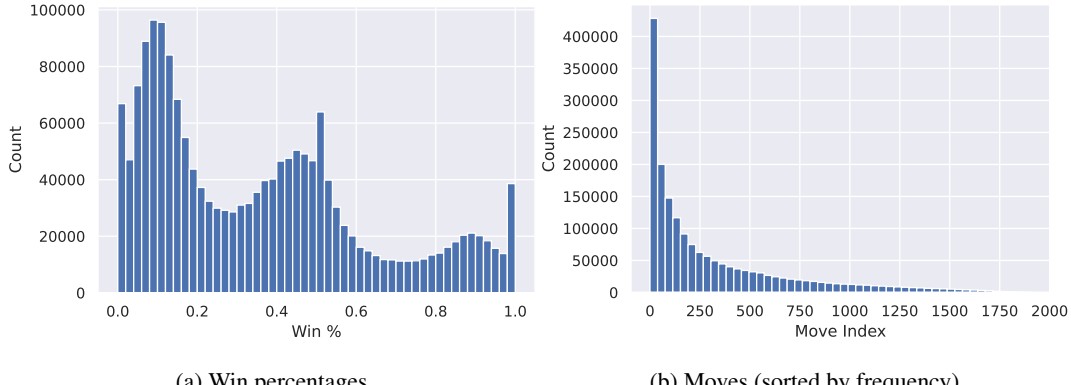

(a) Win percentages           (b) Moves (sorted by frequency)

Figure A1: Win percentage and move distributions for our action-value dataset generated from 1000 games (cf. Table A1). We use 50 buckets to generate the histograms. The win percentage distribution is skewed towards 0 as we consider all legal moves per board and most actions are not advantageous for the player.

played the following numbers of games for the different policies shown in Table 1: 553 games for 9M, 169 games for 136M, 228 games against bots and 174 games against humans for 270M, 181 games for GPT-3.5-turbo-instruct, 37 games for Leela Chess Zero (policy net only), 46 games for Leela Chess Zero (value net only), 44 games for Leela Chess Zero (400 MCTS sim.), 30 games for Stockfish (50ms per move) [oracle], and 42 games for Stockfish (1.5s per board).

## A.6 Stockfish Setup

We use Stockfish 16 (the version from December 2023) throughout the paper. When we play, we use the oracle we used for training, which is an unconventional way to play with this engine: We evaluate each legal move in the position for 50ms, and return the best move based on these scores. This is not entirely equivalent to a standard thinking time of 50ms multiplied by the number of legal moves per position, as we force Stockfish to spend 50ms on moves that could be uninteresting and unexplored. We chose to keep this setup to have a comparison to the oracle we train on. Note that, when comparing the legal moves in a given position, we do not clear Stockfish's cache between the moves. Therefore, due to the way the cache works, this biases the accuracy of Stockfish's evaluation to be weaker for the first moves considered. Finally, due to the nature of our internal hardware setup, we use two different kinds of chips to run Stockfish: (i) to compute the Lichess Elo, we use a 6-core Intel(R) Xeon(R) W-2135 CPU @ 3.70GHz, and (ii) to compute the tournament Elo, we use a single Tensor Processing Unit (V3), as for all the other agents.

## A.7 AlphaZero Setup

We use the AlphaZero [7] version from 2020, with a network trained at that time [23]. We investigate three different versions: (i) policy network only, (ii) value network only and (iii) standard version with search. For (i) we do a search limited to 1 node. For (ii) we do a search limited to 1 node for

every legal action and then take the $\arg\max$. For (iii), we use the standard search from the paper, with 400 MCTS simulations (confirmed via personal correspondence with Schrittwieser et al. [23]) and the exact same UCB scaling parameters. We also take the $\arg\max$ over visit counts. Note that AlphaZero's policy and value network have been trained on 44M games, whereas we trained our largest models on only 10M games.

## A.8 Leela Chess Zero Setup

We use the latest release version (v31.0) of Leela Chess Zero [5] with the largest network officially supported at the time of submission (i.e., the 'Large' T82 network from `https://lczero.org/play/networks/bestnets`). We investigate three different versions: (i) policy network only, (ii) value network only and (iii) standard version with search. For (i) we do a search limited to 1 node. For (ii) we do a search limited to 1 node for every legal action and then take the $\arg\max$. For (iii), we use 400 MCTS simulations. The network architecture and training protocol for T82 were not fully available; more details are now provided in Monroe and Leela Chess Zero Team [6] (written concurrently to our work) about a slightly different architecture, called the ChessFormer.

## A.9 Computational Resources

Our codebase is based on JAX [51] and the DeepMind JAX Ecosystem [52, 53]. We used 4 Tensor Processing Units (V5) per model for the ablation experiments. We used 128 Tensor Processing Units (V5) per model to train our large (9M, 136M and 270M) models. We used a single Tensor Processing Unit (V3) per agent for our Elo tournament.

# B  Additional Results

## B.1  Additional Ablations

We perform additional ablations using the 9M parameter model. As in Section 3.4, the results and conclusions drawn are used to justify our design choices and determine default configuration for model, data, and training. Table A2 shows the results.

**Stockfish time limit**   We create training sets from 1 million games annotated by Stockfish with varying time limits to manipulate the playing strength of our oracle. We report scores on the puzzle set (same for all models) and a test set created using the same time limit as the training set (different for all models). Table 2 shows that a basic time limit of 0.05 seconds gives only marginally worse puzzle performance. As a compromise between computational effort and final model performance we thus choose this as our default value (for our 10M games dataset we need about 15B action-evaluation calls with Stockfish, i.e., roughly 8680 days of unparallelized Stockfish evaluation time).

Table A2: Ablating the Stockfish time limit (used for dataset annotation), the data sampling method for training, and the model architecture (see Appendix B.1).

| Ablation | Parameter | Accuracy (%) | | Kendall's $\tau$ |
|---|---|---|---|---|
| | | **Puzzles** | **Actions** | |
| Stockfish limit [s] | 0.05 | 84.0 | 62.2 | 0.256 |
| | 0.1 | **85.4** | 62.5 | 0.254 |
| | 0.2 | 84.3 | 62.6 | **0.259** |
| | 0.5 | 83.3 | **63.0** | **0.259** |
| Data sampler | Uniform | **83.3** | **63.0** | **0.259** |
| | Weighted | 49.9 | 48.2 | 0.192 |
| Architecture | Transformer | **83.3** | **63.0** | **0.259** |
| | ConvNet | 17.3 | 37.6 | 0.171 |

**Data sampler**   We remove duplicate board states during the generation of the training and test sets. This increases data diversity but introduces distributional shift compared to the "natural" game distribution of boards where early board states and popular openings occur more frequently. To quantify the effect of this shift we use an alternative "weighted" data sampler that draws boards from our filtered training set according to the distribution that would occur if we had not removed duplicates. Results in Table A2 reveal that training on the natural distribution (via the weighted sampler) leads to significantly worse results compared to sampling uniformly randomly from the filtered training set (both trained models are evaluated on a filtered test set with uniform sampling,

Table A3: Inference times and legal move accuracy.

(a) Inference times for the agents from Table 1 on 1000 random baords from the Encyclopedia of Chess Openings (ECO).

| Agent | Inference Time [ms] | |
|---|---|---|
| | Average | Standard Deviation |
| 9M Transformer (ours) | 14.5 | 34.4 |
| 136M Transformer (ours) | 32.0 | 58.5 |
| 270M Transformer (ours) | 57.1 | 117.2 |
| AlphaZero (policy net only) | 4.5 | 4.0 |
| AlphaZero (value net only) | 127.1 | 38.7 |
| AlphaZero (400 MCTS sim.) | 293.1 | 4.4 |
| Leela Chess Zero (policy net only) | 4.3 | 22.5 |
| Leela Chess Zero (value net only) | 116.8 | 51.4 |
| Leela Chess Zero (400 MCTS sim.) | 116.1 | 49.7 |
| Stockfish 16 (50ms per move) [oracle] | 1525.3 | 349.6 |
| Stockfish 16 (1.5s per board) | 1501.1 | 0.6 |

(b) Legal move accuracy on 1000 random boards from three data sources. Measuring the legal move accuracy only really makes sense for BC, which, indeed, almost always plays a legal move. In contrast, our action-value predictor was not trained to play legal moves, since Stockfish's $Q^{\text{SF}}(s, a)$-values are undefined for illegal moves. Note that the legal move accuracy cannot be computed for state-value prediction since it is only defined on boards reachable via legal moves.

| Prediction Target | Legal Move Accuracy (%) | | |
|---|---|---|---|
| | ECO | Puzzles | Test |
| Action-Value | 0.0 | 10.5 | 0.5 |
| Behavioral Cloning | 100.0 | 99.6 | 99.5 |

and the puzzle test set). We hypothesize that the increased performance is due to the increased data diversity seen under uniform sampling. As we train for very few epochs, the starting position and common opening positions are only seen a handful of times during training under uniform sampling, making it unlikely that strong early-game play of our models can be attributed to memorization.

**Model architecture** To investigate the performance of architectures other than transformers on our benchmark, we perform a preliminary investigation with an AlphaZero-like convolutional neural network. We therefore train a residual tower with 8 layers and 256 channels (no pooling, broadcasting, or bottleneck layers), followed by a linear layer and a $\log \text{softmax}$. We input the 8-by-8 board in a single channel and treat the other scalars in the FEN as additional channels. Table A2 shows that this particular model- and training configuration does not lead to strong results for the convolutional architecture we consider. Note that we did not extensively tune our experiment setup and would therefore expect highly domain-specific setups that more closely resemble those of AlphaZero or Leela Chess Zero to achieve much higher performance. Thus, the purpose of this experiment is to show that the vanilla transformer setup seems to achieve better performance than an "equivalently vanilla" convolutional setup.

## B.2    Inference Times

Table A3a shows the inference times in milliseconds on 1000 random boards from the Encyclopedia of Chess Openings [22] using a single TPU (V5). Unsurprisingly, the "searchless" agents (i.e., our models and the policy networks) are orders of magnitudes faster than the agents that rely on tree search at test time. Note that the value networks are slow to evaluate since we compute every legal action individually (and sequentially) and then take the $\arg\max$ over the values.

## B.3    Predicting Legal Moves

A natural question that arises when training models to play chess is whether they have actually learned the rules of the game. However, the problem of generating legal moves is tangential to our paper since we use a legal move generator (given by the environment) for our policies, as is standard in the chess AI literature (see, e.g., AlphaZero, Stockfish, or Leela Chess Zero). In fact, computing the legal move accuracy does not actually make sense for action-value prediction since the action-value function is only defined for legal moves. In contrast, for behavioral cloning the question of legal move accuracy is potentially more interesting (throughout this work, we normalize the predictive distribution to support only the legal moves except when evaluating the legal move accuracy) since the model is trained to output the oracle (and therefore a valid) action.

We investigate the legal move accuracy for action-value and behavioral cloning in Table A3b (the legal move accuracy is undefined for state-value prediction since we can only evaluate boards reachable via legal moves). Unsurprisingly, Table A3b shows that the action-value predictor has a very low legal

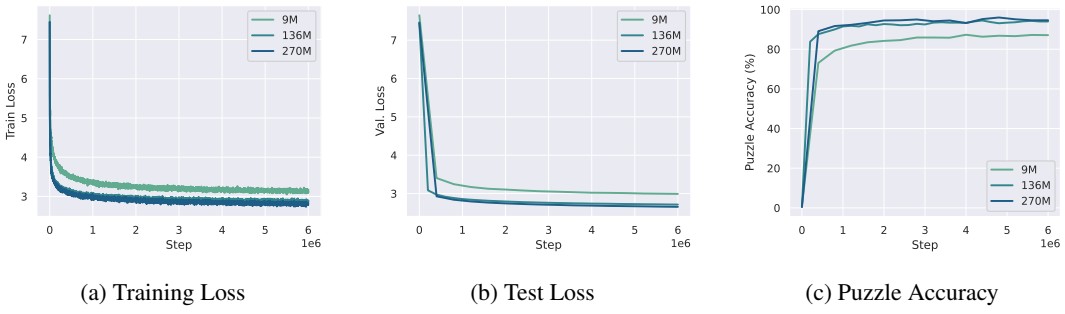

(a) Training Loss     (b) Test Loss     (c) Puzzle Accuracy

Figure A2: Train and test loss curves and puzzle accuracy over time for the models from Section 3.1. We observe no overfitting, which justifies always using the fully trained model in our evaluations.

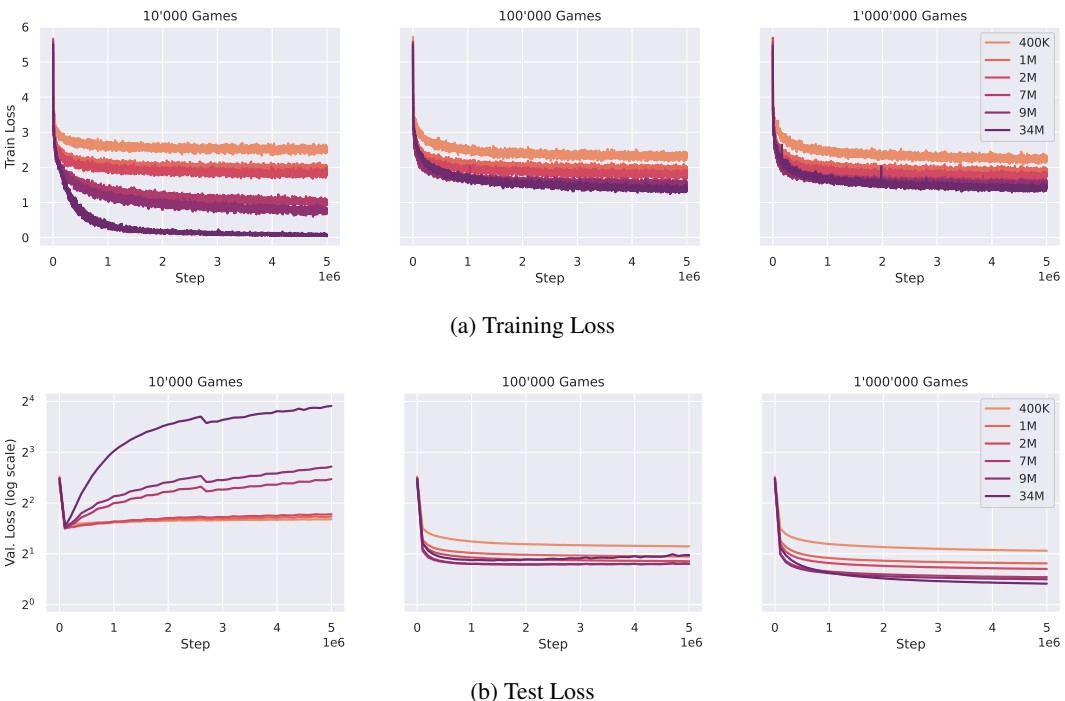

(a) Training Loss

(b) Test Loss

Figure A3: Loss curves when scaling model size and training set size.

move accuracy, since its outputs for illegal moves are undefined (i.e., never adjusted during training). In contrast, our behavioral cloning policy has learned to almost always predict a legal move, even for highly out-of-distribution puzzle boards, showing that our networks have the capacity to learn the rules given our training protocol and dataset – if explicitly trained to do so (i.e., unlike action-value prediction).

## B.4 Loss Curves

In Figure A2 we show the train and test loss curves (and the evolution of the puzzle accuracy) for the large models from Section 3.1. We observe that none of the models overfit and that larger models improve both the training and the test loss.

In Figure A3 we visualize the train and test loss curves for the scaling experiment from Section 3.3. In line with the results shown in the main paper we observe that on the smallest training set, models with $\geq$ 7M parameters start to overfit but not for the larger training sets. Except for the overfitting cases we observe that larger models improve both the training and test loss, regardless of training set size, and that larger training set size improves the test loss when keeping the model size constant.

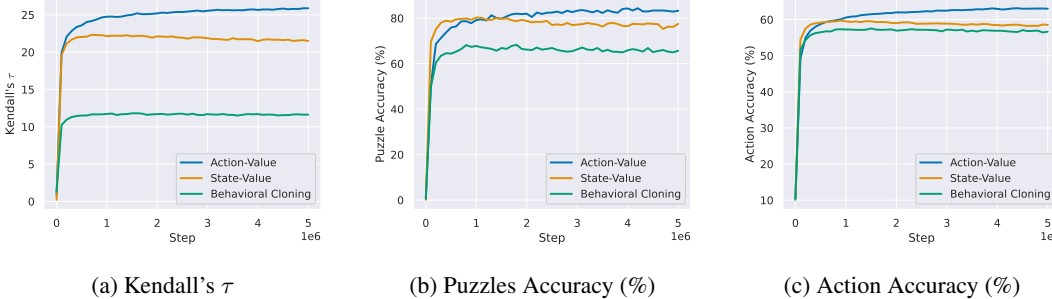

|  |  |  |
|:---:|:---:|:---:|
| (a) Kendall's $\tau$ | (b) Puzzles Accuracy (%) | (c) Action Accuracy (%) |

Figure A4: Comparison of the three different prediction targets (action-value, state-value, and behavioral cloning) trained on the datasets generated from 1 million games. Note that this means that the action-value network is trained on roughly 30 times more data than the other two (cf. Table A1). Action-value learning (trained on 1.6B action-values) learns slightly slower but outperforms the other two variants in the long run (which are trained on roughly 55M states / best actions). Behavioral-cloning falls significantly short of state-value learning, even though both are trained on virtually the same amount of data.

Table A4: Ranking the policies that arise from our three different predictors by having them play against each other in a tournament and computing relative Elo rankings (200 games per pairing; i.e., 600 games per column). When constructing the training data for all three predictors based on the same number of games (middle column), the action-value dataset is much larger than the state-value / behavioral cloning set, which leads to a stronger policy. When correcting for this by forcing the same number of training data points for all three (right column), the difference between state- and action-value prediction disappears.

| | **Relative Tournament Elo** | |
|:---|:---:|:---:|
| **Prediction Target** | **Same # of Games in Dataset** | **Same # of Data Points** |
| Action-Value | **+492** ($\pm 31$) | +252 ($\pm 22$) |
| State-Value | +257 ($\pm 23$) | **+264** ($\pm 22$) |
| Behavioral Cloning | 0 ($\pm 28$) | 0 ($\pm 24$) |

## B.5 Predictor-Target Comparison

In Figure A4 we compare the puzzle accuracy for the three different predictor targets (action-values, state-values, or best action) trained on 1 million games. As discussed in the main text, for a fixed number of games we have very different dataset sizes for state-value prediction (roughly 55 million states) and action-value prediction (roughly 1.6 billion states); see Table A1 for all dataset sizes. It seems plausible that learning action-values might pose a slightly harder learning problem, leading to slightly slower initial learning, but eventually this is compensated for by having much more data to train on compared to state-value learning (see Figure A4, which shows this trend). Also note that since we use the same time-budget per Stockfish call, all action-values for one state use more Stockfish computation time in total (due to one call per action) when compared to state-values (one call per board). To control for the effect of dataset size, we train all three predictors (9M parameter model) on a fixed set of 40 million data points. Results are shown in Figure A5. As the results show, the state-value policy in this case slightly outperforms the action-value policy, except for action-ranking (Kendall's $\tau$), which makes sense since the action-value predictor is implicitly trained to produce good action rankings. To see how this translates into playing-strength, we pit all three policies (AV, SV, BC) against each other and determine their relative Elo rankings. Table A4 shows that when not controlling for the number of training data points, the action-value policy is strongest (in line with the findings in Table 2 and Figure A4), but when controlling for the number of training data points the action-value and state-value policy perform almost identical (in line with Figure A5).

Throughout all these results we observe lower performance of the behavioral cloning policy, despite being trained on a comparable number of data points as the state-value policy. The main hypothesis for this is that the amount of information in the behavioral cloning dataset is lower than the state-value

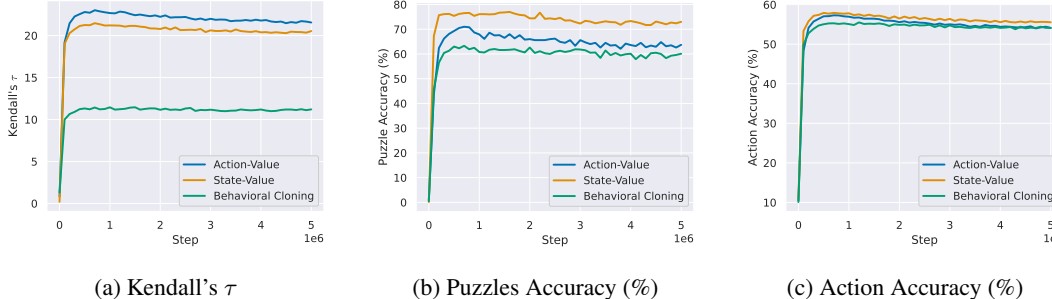

| (a) Kendall's $\tau$ | (b) Puzzles Accuracy (%) | (c) Action Accuracy (%) |

Figure A5: Comparison of the three different prediction targets (action-value, state-value, and behavioral cloning) trained on exactly the same number of data points (40M). The superiority of action-value learning over state-value learning disappears (or even reverses to some degree), except when measuring the action-ranking correlation (Kendall's $\tau$) which the action-value policy is indirectly trained to perform well on.

dataset, since we throw away any information in the state- or action-values beyond the index of the oracle action. We suspect that training on the full action distribution of the oracle (with cross-entropy loss), rather than the best action only would largely close this gap, but we consider this question beyond the scope of this paper and limit ourselves to simply reporting the observed effect in our setting.

### B.6 Fischer Random Chess

As a measure of out-of-distribution generalization, we evaluate the performance of our 9M transformer on Fischer random chess[3] (Chess960) via Lichess (using the same setup as for Table 1). Fischer is a variation of the game of chess, which employs the same board and pieces as classical chess, but the starting position of the pieces on the players' home ranks is randomized, following certain rules. The random setup makes gaining an advantage through the memorization of openings impracticable; players instead must rely more on their skill and creativity over the board. Thus, Fischer random chess can be viewed as a measure of whether our models have learned general tactics (for the early stages of the game). While our bot attains a blitz Elo of 2054 (cf. Table 1), it only manages to play Fischer random chess at 1539 Elo (against bots), suggesting that the distribution shift is too high for our models to handle.

### B.7 Tactics

In Figure A6, we analyze the tactics learned by our 270M transformer used against a human with a blitz Elo of 2145. We observe that our model has learned to sacrifice material when it is advantageous to build a longer-term advantage.

### B.8 Playing Style

We recruited chess players of National Master level and above to analyze our agent's games against bots and humans on the Lichess platform. They made the following qualitative assessments of its playing style and highlighted specific examples (see Figure A7).

The agent has an aggressive enterprising style where it frequently sacrifices material for long-term strategic gain. The agent plays optimistically: it prefers moves that give opponents difficult decisions to make even if they are not always objectively correct. It values king safety highly in that it only reluctantly exposes its own king to danger but also frequently sacrifices material and time to expose the opponent's king. For example 17 .. Bg5 in game B.8.1 encouraged its opponent to weaken their king position. Its style incorporates strategic motifs employed by the most recent neural engines [7, 54]. For example it pushes wing pawns in the middlegame when conditions permit (see game B.8.2). In game B.8.3 the agent executes a correct long-term exchange sacrifice. In game B.8.4 the bot uses a

---

[3] https://en.wikipedia.org/wiki/Fischer_random_chess

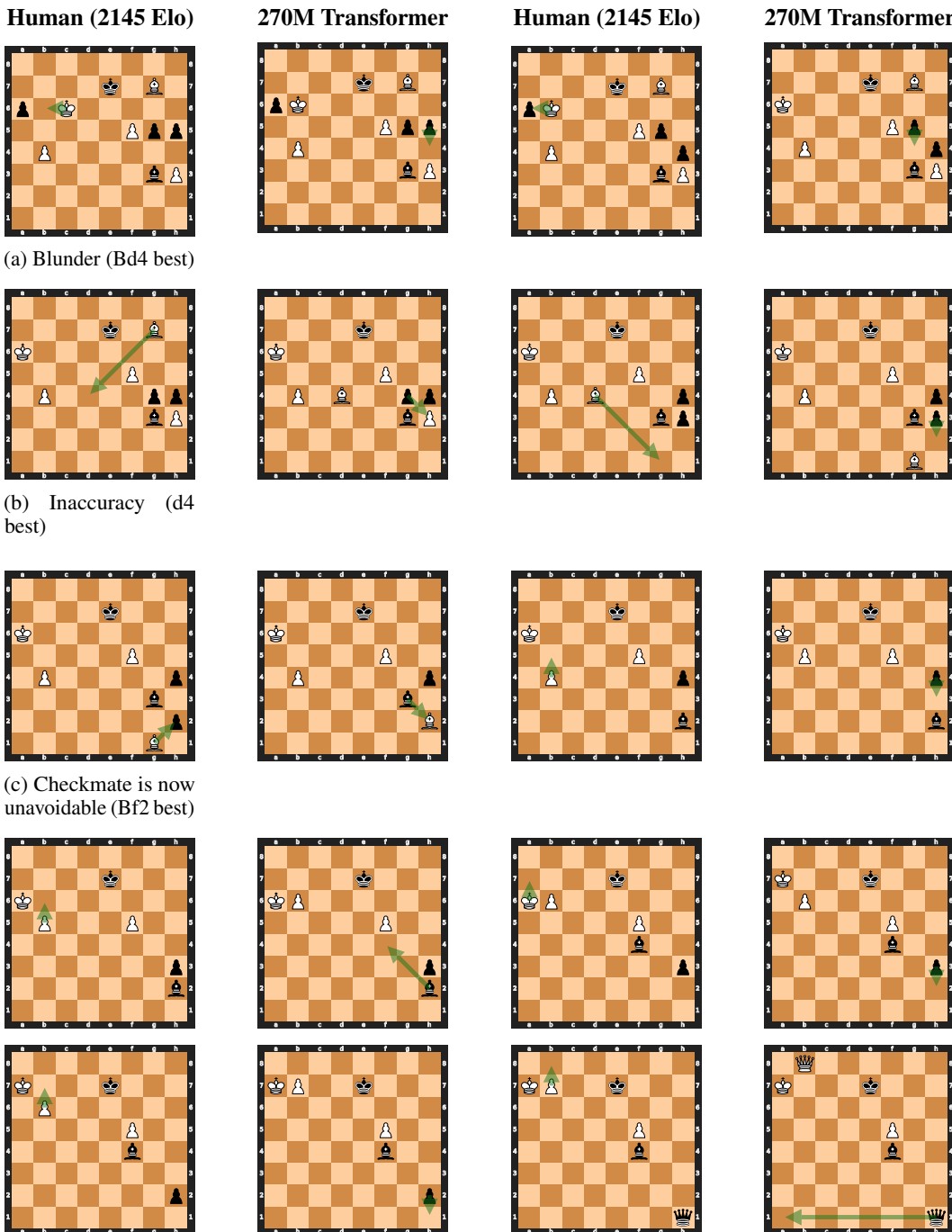

Figure A6: Example of the learned tactics for our 270M transformer vs. a human player with a blitz Elo of 2145 (the game progresses from left to right and top to bottom). Our model decides to sacrifice two pawns since the white bishop will not be able to prevent it from promoting one of the pawns. The individual subfigure captions contain the Stockfish analysis from Lichess (i.e., our model plays optimally).

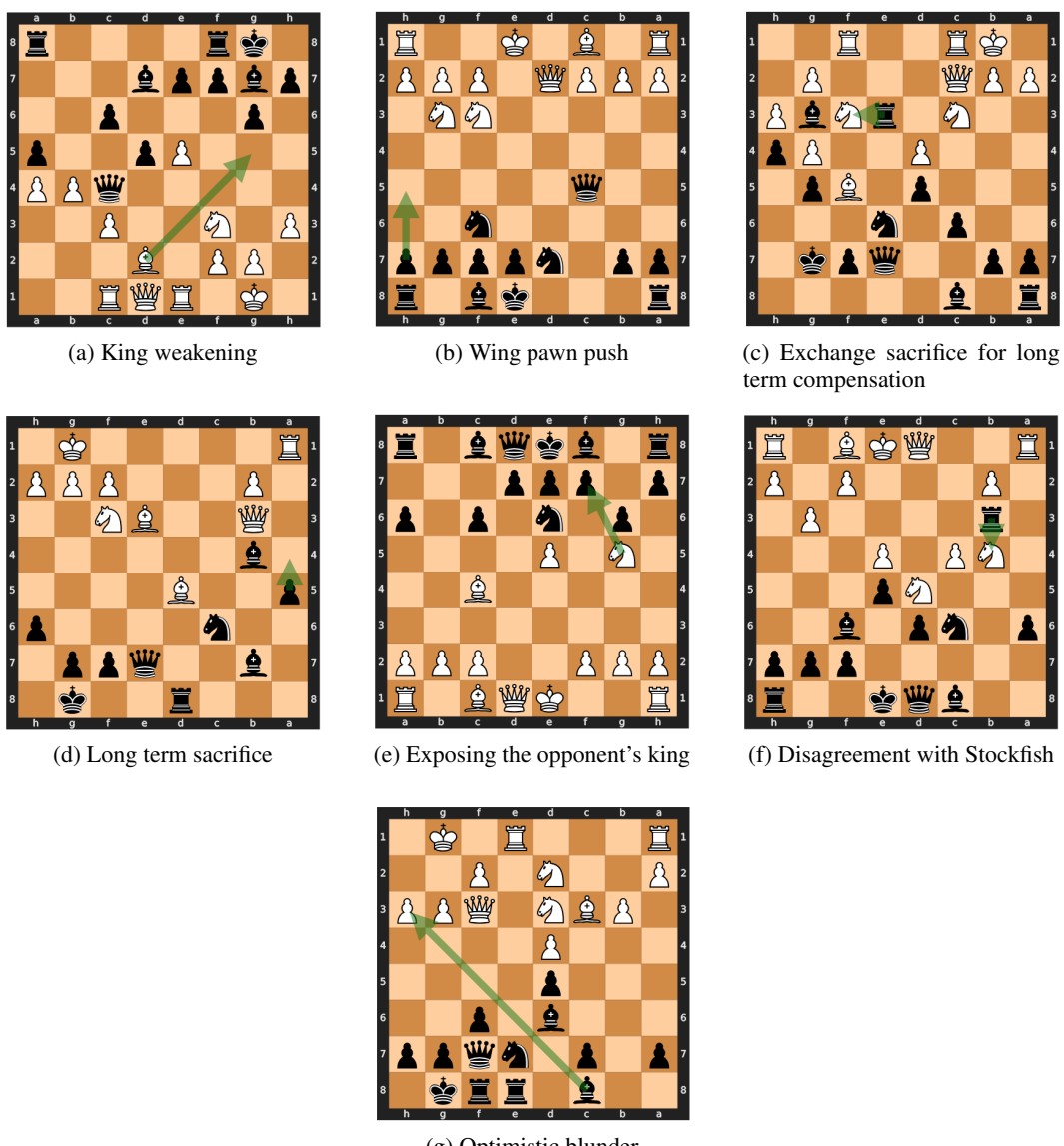

(a) King weakening

(b) Wing pawn push

(c) Exchange sacrifice for long term compensation

(d) Long term sacrifice

(e) Exposing the opponent's king

(f) Disagreement with Stockfish

(g) Optimistic blunder

Figure A7: Examples of our 270M transformer's playing style against online human opponents.

motif of a pin on the back rank to justify a pawn sacrifice for long term pressure. Game B.8.5 features a piece sacrifice to expose its opponent's king. The sacrifice is not justified according to Stockfish although the opponent does not manage to tread the fine line to a permanent advantage and blunders six moves later with Bg7.

The agent has a distinct playing style to Stockfish: one analyzer commented "it feels more enjoyable than playing a normal engine", "as if you are not just hopelessly crushed". Indeed it does frequently agree with Stockfish's move choices suggesting that the agent's action-value predictions match Stockfish's. However the disagreements can be telling: the piece sacrifice in the preceding paragraph is such an example. Also, game B.8.6 is interesting because the agent makes moves that Stockfish strongly disagrees with. In particular the agent strongly favours 18 .. Rxb4 and believes black is better, in contrast Stockfish believes white is better and prefers Nd4. Subsequent analysis by the masters suggests Stockfish is objectively correct in this instance. Indeed on the very next move the agent has reversed its opinion and agrees with Stockfish.

The agent's aggressive style is highly successful against human opponents and achieves a grandmaster-level Lichess blitz Elo of 2895. However, against other engines its estimated Elo is far lower, i.e., 2299. Its aggressive playing style does not work as well against engines that are adept at tactical calculations, particularly when there is a tactical refutation to a sub-optimal move. Most losses against bots can be explained by just one tactical blunder in the game that the opponent refutes. For example Bxh3 in game B.8.7 loses a piece to g4.

Finally, the recruited chess masters commented that the agent's style makes it very useful for opening repertoire preparation. It is no longer feasible to surprise human opponents with opening novelties as all the best moves have been heavily over-analyzed. Modern opening preparation amongst professional chess players now focuses on discovering sub-optimal moves that pose difficult problems for opponents. This aligns extremely well with the agent's aggressive, enterprising playing style which does not always respect objective evaluations of positions.

### B.8.1  King weakening game

1. e4 c5 2. Nf3 Nc6 3. Bb5 g6 4. O-O Bg7 5. c3 Nf6 6. Re1 O-O 7. d4 d5 8. e5 Ne4 9. Bxc6 bxc6 10. Nbd2 Nxd2 11. Bxd2 Qb6 12. dxc5 Qxc5 13. h3 Qb5 14. b4 a5 15. a4 Qc4 16. Rc1 Bd7 17. Bg5 f6 18. Bd2 Bf5 19. exf6 exf6 20. Nd4 Bd7 21. Nb3 axb4 22. cxb4 Qh4 23. Nc5 Bf5 24. Ne6 Rfc8 25. Nxg7 Kxg7 26. Re7+ Kh8 27. a5 Re8 28. Qe2 Be4 29. Rxe8+ Rxe8 30. f3 1-0

### B.8.2  Wing pawn push game

1. e4 c6 2. d4 d5 3. Nc3 dxe4 4. Nxe4 Nf6 5. Ng3 c5 6. Bb5+ Bd7 7. Bxd7+ Nbxd7 8. dxc5 Qa5+ 9. Qd2 Qxc5 10. Nf3 h5 11. O-O h4 12. Ne2 h3 13. g3 e5 14. Nc3 Qc6 15. Qe2 Bb4 16. Bd2 O-O 17. Rae1 Rfe8 18. Ne4 Bxd2 19. Qxd2 Nxe4 0-1

### B.8.3  Exchange sacrifice game

1. d4 d5 2. c4 e6 3. Nc3 Bb4 4. cxd5 exd5 5. Nf3 Nf6 6. Bg5 h6 7. Bh4 g5 8. Bg3 Ne4 9. Rc1 h5 10. h3 Nxg3 11. fxg3 c6 12. e3 Bd6 13. Kf2 h4 14. g4 Bg3+ 15. Ke2 O-O 16. Kd2 Re8 17. Bd3 Nd7 18. Kc2 Rxe3 19. Kb1 Qe7 20. Qc2 Nf8 21. Rhf1 Ne6 22. Bh7+ Kg7 23. Bf5 Rxf3 24. gxf3 Nxd4 25. Qd3 Nxf5 26. gxf5 Qe5 27. Ka1 Bxf5 28. Qe2 Re8 29. Qxe5+ Rxe5 30. Rfd1 Bxh3 31. Rc2 Re3 32. Ne2 Bf5 33. Rcd2 Rxf3 34. Nxg3 hxg3 0-1

### B.8.4  Long term sacrifice game

1. d4 d5 2. c4 e6 3. Nf3 Nf6 4. Nc3 Bb4 5. Bg5 dxc4 6. e4 b5 7. a4 Bb7 8. axb5 Bxe4 9. Bxc4 h6 10. Bd2 Bb7 11. O-O O-O 12. Be3 c6 13. bxc6 Nxc6 14. Qb3 Qe7 15. Ra4 a5 16. Rd1 Rfd8 17. d5 exd5 18. Nxd5 Nxd5 19. Rxd5 Rxd5 20. Bxd5 Rd8 21. Ra1 a4 22. Rxa4 Qd7 23. Bc4 Qd1+ 24. Qxd1 Rxd1+ 25. Bf1 Ba5 26. Rc4 Rb1 27. Rc2 Nb4 28. Rc5 Nc6 29. Bc1 Bb4 30. Rc2 g5 31. h4 g4 32. Nh2 h5 33. Bd3 Ra1 34. Nf1 Ne5 35. Be2 Be4 36. Rc8+ Kh7 37. Be3 Re1 38. Bb5 Bd3 39. Bxd3+ Nxd3 40. Rd8 Nxb2 41. Rd5 Be7 42. Rd7 Bxh4 43. g3 Bf6 44. Rxf7+ Kg6 45. Rxf6+ Kxf6 46. Bd4+ Kg5 47. Bxb2 Rb1 48. Bc3 Kf5 49. Kg2 Rb3 50. Ne3+ Ke4 51. Bf6 Rb5 52. Kf1 Rb6 53. Bc3 Rb3 54. Bd2 Kd3 55. Be1 Rb5 56. Ng2 Ke4 57. Ke2 Rb2+ 58. Bd2 Rc2 59. Ne3 Ra2 60. Nc4 Kd4 61. Nd6 Ke5 62. Ne8 Kf5 63. Kd3 Ra6 64. Bc3 Rc6 65. Bb4 Kg6 66. Nd6 Ra6 67. Bc5 Ra5 68. Bd4 Ra6 69. Nc4 Ra4 70. Nb6 Ra5 71. Ke4 h4 72. gxh4 Kh5 73. Bf6 Ra2 74. Ke3 Ra3+ 75. Ke2 g3 76. Nd5 Ra2+ 77. Kf3 gxf2 78. Nf4+ Kh6 79. Kg2 f1=Q+ 80. Kxf1 Rc2 81. Bg5+ Kh7 82. Ne2 Kg6 83. Kf2 Ra2 84. Kf3 Ra4 85. Ng3 Rc4 86. Bf4 Rc3+ 87. Kg4 Rc4 88. h5+ Kf6 89. Nf5 Ra4 90. Ne3 Ra5 91. Nc4 Ra4 92. Ne5 Kg7 93. Kf5 Ra5 94. Kg5 Rb5 95. Kg4 Rb1 96. Kf5 Rb5 97. Ke4 Ra5 98. h6+ Kh7 99. Bd2 Ra2 100. Be3 Ra6 101. Ng4 Ra3 102. Bd2 Ra2 103. Bf4 Ra5 104. Kf3 Rf5 105. Ke3 Kg6 106. Ke4 Rh5 107. Kf3 Rh3+ 108. Kg2 Rh5 109. Kg3 Ra5 110. Be3 Ra3 111. Kf3 Rb3 112. Ke4 Rb4+ 113. Bd4 Ra4 114. Ke5 Rc4 115. Kd5 Ra4 116. Ke4 Rb4 117. Kd3 Ra4 118. Kc3 Ra3+ 119. Kc4 Rg3 120. Ne3 Rh3 121. Kd5 Rxh6 122. Bb6 Rh3 123. Nc4 Rh5+ 124. Ke6 Rg5 125. Nd2 Rg2 126. Nf1 Rb2 127. Bd8 Re2+ 128. Kd5 Re1 129. Ne3 Rxe3 130. Bh4 Kf5 131. Bf2 Rd3+ 132. Kc4 Ke4 133. Bc5 Rc3+ 134. Kxc3 1/2-1/2

### B.8.5  Expose king game

1. e4 c5 2. Nf3 Nc6 3. Na3 Nf6 4. e5 Nd5 5. d4 cxd4 6. Nb5 a6 7. Nbxd4 g6 8. Bc4 Nc7 9. Nxc6 bxc6 10. Ng5 Ne6 11. Nxf7 Kxf7 12. Bxe6+ Kxe6 13. Bd2 Kf7 14. Qf3+ Kg8 15. e6 dxe6 16.

O-O-O Qd5 17. Qe3 Bg7 18. Bc3 Qxa2 19. Rd8+ Kf7 20. Qf4+ Bf6 21. Rxh8 Qa1+ 22. Kd2 Qxh1 23. Bxf6 exf6 24. Qc7+ 1-0

## B.8.6 Stockfish disagreement game

1. e4 c5 2. Nf3 Nc6 3. d4 cxd4 4. Nxd4 Nf6 5. Nc3 e6 6. Ndb5 d6 7. Bf4 e5 8. Bg5 a6 9. Na3 b5 10. Nd5 Qa5+ 11. Bd2 Qd8 12. Bg5 Be7 13. Bxf6 Bxf6 14. c4 b4 15. Nc2 Rb8 16. g3 b3 17. axb3 Rxb3 18. Ncb4 Rxb4 19. Nxb4 Nxb4 20. Qa4+ Kf8 21. Qxb4 g6 22. Bg2 h5 23. h4 Kg7 24. O-O g5 25. hxg5 Bxg5 26. f4 Be7 27. fxe5 dxe5 28. Qc3 Bc5+ 29. Kh2 Qg5 30. Rf5 Bxf5 31. Qxe5+ Qf6 32. Qxf6+ Kxf6 33. exf5 Kg5 34. Bd5 Rb8 35. Ra2 f6 36. Be6 Kg4 37. Kg2 Rb3 38. Bf7 Rxg3+ 39. Kf1 h4 40. Ra5 Bd4 41. b4 h3 42. Bd5 h2 43. Bg2 Rb3 44. Rxa6 Rb1+ 45. Ke2 Rb2+ 0-1

## B.8.7 Blunder game

1. b3 e5 2. Bb2 Nc6 3. e3 d5 4. Bb5 Bd6 5. Bxc6+ bxc6 6. d3 Qg5 7. Nf3 Qe7 8. c4 Nh6 9. Nbd2 O-O 10. c5 Bxc5 11. Nxe5 Bb7 12. d4 Bd6 13. O-O c5 14. Qh5 cxd4 15. exd4 Rae8 16. Rfe1 f6 17. Nd3 Qf7 18. Qf3 Bc8 19. h3 Nf5 20. g3 Ne7 21. Bc3 Bxh3 22. g4 f5 23. Qxh3 fxg4 24. Qxg4 h5 25. Qe6 g5 26. Qxf7+ Rxf7 27. Bb4 Ref8 28. Bxd6 cxd6 29. b4 Nf5 30. Re6 Kg7 31. Rd1 Rc7 32. Nf3 g4 33. Nd2 h4 34. Nb3 Rc2 35. Nf4 g3 36. Nh5+ Kh7 37. fxg3 Nxg3 38. Nxg3 Rg8 39. Rd3 Rxa2 40. Rxd6 Rb2 41. Rxd5 Rxg3+ 42. Rxg3 hxg3 43. Nc5 Kg6 44. b5 Rxb5 45. Kg2 a5 46. Kxg3 a4 47. Rd6+ Kf5 48. Nxa4 Rb3+ 49. Kf2 Rh3 50. Nc5 Kg5 51. Rc6 Kf5 52. d5 Ke5 53. d6 Rh2+ 54. Kg3 Rd2 55. d7 Rxd7 56. Nxd7+ Ke4 57. Rd6 Ke3 58. Nf6 Ke2 59. Ng4 Ke1 60. Kf3 Kf1 61. Rd1# 1-0

