# OpenReview forum: "Amortized Planning with Large-Scale Transformers: A Case Study on Chess"
_NeurIPS.cc/2024/Conference — NeurIPS 2024 poster_

### Official Review · Reviewer_tXJ6 · 2024-07-11

**Soundness:** 4
**Presentation:** 4
**Contribution:** 3
**Rating:** 7
**Confidence:** 3

**Summary:**

This paper introduces an open dataset comprised of real chess positions annotated with a evaluation score, the best move to play, and a score for each legal move according to StockFish 16 using 50ms for each action. They then train large transformers over this dataset, and include in-depth analysis through making the model play on Lichess, and using ablation studies over the hyperparameters of the network.

**Strengths:**

- The paper provides an extensive analysis of the current state and abilities of their trained Transformer on ChessBench, giving a really strong starting point for future work on the subject.
- The promising results of using a Transformer architecture to replicate StockFish's evaluation point to the ability of large models to be able to perform well in the area of planning for complex tasks.
- The dataset introduced by the authors is quite extensive, and could prove very useful for future research

**Weaknesses:**

- The dataset might be of relatively poor quality, as it uses Stockfish 16 with only 50ms for each evaluation. This could make a significant proportion of the evaluations be wrong, and potentially limit the playing strength of models trained using this data. (Table 1. show a significant (>200 elo) disparity in playing strength between 0.05s and 1.5s for stockfish)
- The goal of the paper is not immediately obvious. A first reading of the abstract and the introduction seem to imply with the title that the paper is concerned about the general abilities of Transformers to tackle planning problems, while the contributions are more focused on providing a dataset (as well as an initial case study on it) for future research on this topic, focused on the case of chess.

**Questions:**

Section 2.2 mentions that the problem was seen as a classification problem, binning the win percentages into K uniform bins. Was augmenting the density of bins around 50% in order to improve accuracy in critical positions considered?

Table A3(a) in the appendix seems to show that stockfish with 0.05s per move takes 1.5s per move on average. What does this mean? Wasn't the 50ms limit a strict limit?

Appendix A9 mentions the hardware used for the experiments, but not how long it took to train each model. Would it be possible to add this information?

**Limitations:**

The limitations are adequately addressed.

---

> ### Author Rebuttal · Authors · 2024-08-06
>
> We thank the reviewer for their insightful and positive feedback.
>
>
> **Is the dataset of poor quality if you use Stockfish with only 50ms evaluation time (as there is a significant Elo disparity between 0.05s and 1.5s for Stockfish)? Also, why does Stockfish with 0.05s evaluation time take 1.5s to play a move on average?**
>
> The dataset is of high (super grandmaster) quality, as evidenced by the fact that Stockfish with 0.05s, i.e., the data-generating oracle, achieves an Elo of 2713 against other bots on Lichess. While higher time limits would lead to even stronger annotations, there is a trade-off in terms of computing time spent on collecting the dataset (currently 8680 days of unparallelized compute). Note that we did, however, create smaller-scale datasets with higher annotation time limits in Table A2, showing that none of our networks can currently fully match the performance of the data-generating oracle(s), so the current frontier seems to lie with improving models rather than the playing-strength of the data-generating oracle.
>
> To clarify the 0.05s vs. 1.5s evaluation time, we consider two versions of Stockfish:
> * *Data-generating oracle*: For a given board, the oracle *evaluates* every legal move separately for 0.05s (i.e., if there are, e.g., 30 legal moves, Stockfish uses a total time of 1.5s to *play* a move). We primarily compare to this version since this is how we constructed the dataset.
> * *“Standard-play” Stockfish*: For completeness, we also include Stockfish as it would typically be used in standard play, i.e., its evaluation time is restricted per state but not per move, meaning that some clearly suboptimal moves can receive very little time (based on Stockfish’s pruning) and the resulting extra budget is instead spent on more promising moves. Since there are roughly 30 legal moves per board on average, we choose a time limit of 30 * 0.05s = 1.5s per board state for  “standard-play” Stockfish. This leads to an improvement in play of >200 Elo compared to the version that spends a fixed 50ms on each legal move per board state, but we only include this baseline as a “reference” (i.e., it is not reflected in the dataset).
>
> We agree that the difference between these two versions is subtle, and we will clarify it in the next revision of our paper.
>
>
> **How long did it take to train each model?**
>
> The 9M models trained at roughly 270 steps per second, yielding a total training time of 10M / 270 * 3600) = 10.2 hours. The 136M models trained at approximately 26 steps per second, yielding a total training time of 10M / (26 * 3600 * 24) = 4.45 days. The 270M models trained at roughly 13 steps per second, yielding a total training time of 10M / (13 * 3600 * 24) = 8.9 days. We will add these details to Appendix A.6, which describes our hardware setup.
>
>
> **Did you consider non-uniform binning to improve accuracy in critical positions?**
>
> We initially experimented with non-uniform binning but quickly abandoned the approach due to its increased complexity (e.g., state-value expansion is non-trivial with non-uniform bins due to its minimax nature) and failure to produce large performance gains. However, we did ablate the number of bins in Table A2, showing that an increased resolution improves performance, but only up to a certain point. Based on our experience, non-uniform binning can lead to gains when the overall number of bins is very small, but these gains diminish relatively rapidly with medium to large numbers of bins.

---

> > ### Comment · Reviewer_tXJ6 · 2024-08-10
> >
> > I acknowledge that I have read the rebuttal. The revisions and clarifications are welcome, and I will maintain my evaluation.

---

### Official Review · Reviewer_Ye4U · 2024-07-12

**Soundness:** 3
**Presentation:** 3
**Contribution:** 2
**Rating:** 6
**Confidence:** 3

**Summary:**

This paper aims to solve the challenging problem in the chess game. The main contributions include building a large-scale dataset and training a large-scale transformer model with the collected dataset. The proposed approach has shown a significant outperformance over the baselines.

**Strengths:**

The writing is mostly clear.

The empirical performance shows a great improvement.

Chess is a challenging game to evaluate the level of artificial intelligence, and the collected dataset can be used for the following research works.

**Weaknesses:**

The technique novelty of this work is incremental, as the techniques used in the proposed approach have been previously developed. The usage of transformer in amortized planning is a contribution, but not significant enough.

**Questions:**

Can the authors elaborate on whether the transformer structure is more important or the dataset is more important for performance improvement?

**Limitations:**

The authors have discussed the potential limitations.

---

> ### Author Rebuttal · Authors · 2024-08-06
>
> We thank the reviewer for their helpful feedback.
>
>
> **Is the transformer or the dataset more important for performance improvement?**
>
> Although this cannot be determined with certainty given our results, we investigated this question in Table A2, where we ablated model architecture and the time limit used when annotating a move with Stockfish for the dataset creation. Table A2 shows that using a higher time limit does lead to some performance gains (e.g., 1.4% higher puzzle accuracy) but at an extremely high computational cost (i.e., doubling the total unparalleled annotation time from 8680 days to 17360 days). Therefore, we compromise between computational effort and final model performance and use 0.05s per state-action value annotation.
>
>
> **Using transformers for amortized planning is a contribution, but not significant enough.**
>
> We respectfully disagree. Though the approach of amortizing a planning algorithm with a fixed-parametric function approximator may be evident to some researchers, it is typically not the mainstream interpretation. Large sequence models have been repeatedly dismissed as mere “statistical pattern matchers”,  “stochastic parrots”, or “curve fitters”; implying that this approach is insufficient to capture complex algorithmic behavior. While the results of large foundation models may speak for themselves w.r.t. this criticism, we believe it is important to advocate and thoroughly test the amortization viewpoint. Just to reiterate: the state space of chess is very large - even when only considering the “natural” distribution of board states from games on lichess.org, all but the most trivial games face our networks with mostly unseen board states, where accurate value estimates and/or actions require considerable generalization beyond the training data. Our results cannot be explained as memorization with a bit of interpolation or simple statistical pattern matching, and not too long ago, such results were thought to be unachievable without explicit planning. Our results also show that transformers (up to the size used in our experiments) cannot always fully match Stockfish. Together, this raises two important future questions: (i) What is missing to match Stockfish’s performance; is it just a matter of scale, or can we identify systematic shortcomings of transformers? (ii) How do transformers implement amortized planning, and how is it related to architectural parameters such as depth? While these questions are beyond the scope of our current work, we believe that our results and dataset lay excellent groundwork to investigate these questions in the future.

---

### Official Review · Reviewer_LZwX · 2024-07-13

**Soundness:** 4
**Presentation:** 4
**Contribution:** 2
**Rating:** 6
**Confidence:** 5

**Summary:**

The paper introduces a large dataset of chess board states along with annotations for best moves and state-(action)-values. It demonstrates that transformers trained on this dataset with supervised learning can achieve significant performance and generalization. This adds to the growing evidence showing that neural nets can implement complex behaviors (such as imitating search-based chess engines).

**Strengths:**

- There is continuing debate about the extent to which neural networks learn generalizing non-trivial algorithms, so this is a timely addition.
- The paper has extensive experiments with a wealth of different metrics and some interesting ablations.
- I like the discussion in section 4 and elsewhere. It's transparent about potential issues, but in my view none of these are a big issue for the overall takeaways, and it seems fine to add manual workarounds to deal with them (like the authors do).
- The writing is very clear and covers all the details I was interested in. Overall, I really like the execution of this paper.

**Weaknesses:**

EDIT: see my reply (https://openreview.net/forum?id=XlpipUGygX&noteId=UsMmll2N3a) for my updated stance.

The paper does not add too much that didn't already exist outside peer-reviewed ML venues. As the authors discuss (line 312), the existing Leela Chess Zero network is stronger than the network introduced by this paper. The T82 version of Leela (which this paper compares against) is also trained using supervised learning and uses a transformer architecture similar to the one in this paper. Similarly, datasets of chess board states with engine annotations exist (e.g. at https://database.lichess.org/). I'm not aware of datasets with state-action-value annotations for all moves in a state, but it's not clear that this is even necessary to train a state-action-value predictor.

In my mind, the main contribution of this paper is thus creating a well-documented and peer-reviewed chess-playing neural network and dataset, as well as running ablations (which likely don't come as a surprise to the chess engine community, but seem interesting for the broader NeurIPS audience). This is still a nice contribution (though I wish it was made clearer from the start that this is what the paper does, e.g., in the intro/"contributions" paragraph. EDIT: the authors' changes do make this clear now).

Minor notes:
- Regarding why BC does worse than value predictions (appendix B.5): I don't think the richer training signal is the full explanation, since the value nets of Leela/AlphaZero also outperform their policy nets, even though those policy nets are trained to imitate a full distribution (rather than the single best move). My guess is that another reason is the weak form of "1-ply search" inherent in letting the network evaluate many different actions/follow-up states and taking an argmax.
- Line 177: T82 is a transformer, not a ConvNet (see more below)
- Nitpick for table 1 and elsewhere: Leela and AlphaZero get the past eight board states as input, rather than the full move history (though the lack of even earlier states shouldn't matter much). I think the most notable difference is in fact between GPT-3.5 vs all the other networks, since GPT-3.5 gets *only* moves and so has learn to keep track of the board state itself. So in the "Input" column, I might distinguish between "PGN" (for GPT-3.5), "FEN" (for the new models), and "FEN + past states" (for Leela and AlphaZero).
- On the claim that the network plays at grandmaster level: this is a difficult comparison to make, since human players play better when given more time per move, whereas the neural network of course doesn't improve from time beyond what's needed for one or a fixed number of forward passes. I agree that the network seems as strong as grandmasters with Blitz time controls, but it might still be significantly weaker than grandmasters at classical time controls. I think this distinction isn't always clear (e.g. line 44).

Detailed notes on Leela vs the new model in this paper: The T82 version of Leela that this paper compares against was trained using supervised learning on MCTS data, rather than trained with online RL (this isn't well-documented, but see https://lczero.org/play/networks/basics/ and note that T82 is one of the "contrib runs"). It is also a transformer with an input encoding not too different from the one used in this paper (one token position per square of the chess board). The main differences as far as I can tell are that Leela has some domain-specific improvements to its architecture (https://lczero.org/blog/2024/02/transformer-progress/, Smolgen is part of T82), uses fewer parameters, and was trained on different (and probably more) data.

**Questions:**

(pretty minor)
- If I understand correctly, all three types of predictors use entirely separate models. I'm wondering whether training a single model with a shared body and three small prediction heads would lead to transfer between predictors (e.g. let the BC policy profit from the larger amount of data used to train the state-action-value predictor). Is this something you considered/tried?
- Why is the state-value predictor trained only on the smaller amount of data (compared to the state-action-value predictor)? Couldn't you use the state-action-value annotations as state-value annotations for the follow-up state, which would mean you'd already have all the training data needed to also train the state-value predictor on more data?

**Limitations:**

Issues are discussed well in the paper

---

> ### Author Rebuttal · Authors · 2024-08-06
>
> We thank the reviewer for their careful study of our paper and, in particular, for all their clarifying remarks on lc0, which have helped us to improve the description of that baseline.
>
>
> **Lc0’s T82 network is a domain-specific transformer (rather than a ConvNet), trained using supervised learning on an unknown amount of MCTS data (rather than using online RL).**
>
> Thank you for these clarifications and all the additional information on lc0! We agree with the reviewer that these details are not well-documented, which makes a scientific comparison beyond evaluating playing strength difficult (e.g., it is unclear which and how much data was used to train T82). We have updated the description of lc0 in Section 2.5 as follows:
>
> “**Leela Chess Zero** We consider three variants: (i) with 400 MCTS simulations, (ii) only the policy network, and (iii) only value network (where (ii) and (iii) perform no additional search). We use the T82 network, which is a transformer and the largest network available on the official Leela Chess Zero website [5]. T82 uses one token per square of the chess board, has fewer parameters (94M vs. up to 270M for our models), includes some domain-specific architecture changes (e.g., Smolgen [29]), and was trained using supervised learning on MCTS data.”
>
>
> **The grandmaster claim against humans may only hold with Blitz time controls (since human performance improves with more evaluation time, unlike the models trained in this paper), which isn’t always clear in the paper.**
>
> We fully agree that our results only imply grandmaster-level performance against humans with Blitz time controls, and we will clarify this distinction throughout the paper. For example, we have changed line 44 to “... playing chess at a high level (grandmaster) against humans *with Blitz time controls*”.
>
>
> **BC does worse than value predictions because it performs a weak form of “1-ply search” rather than having a poorer training signal (lc0’s/AZ’s policy nets are also worse even though they imitate the full distribution, i.e., not just the best move).**
>
> This is an interesting observation. Indeed, multiple factors probably contribute to BC’s relatively weak performance, and we agree that inference-time FLOPS (i.e., performing a less expansive “1-ply search” than the value-based methods) is probably also a major factor. We have expanded the discussion of these results in Appendix B5 accordingly.
>
>
> **AZ and lc0 operate on the past 8 board states rather than the full move history.**
>
> Thank you for the clarification! We agree that the reviewer’s proposed distinction of “PGN” for GPT-3.5 and “FEN + past states” for AZ and lc0 is more precise and have updated Table 1 and the description in Section 3.1 accordingly.
>
>
> **Did you try training a single model with three prediction heads for the different predictor targets (AV, BC, SV) to enable transfer between them (e.g., more data for BC)?**
>
> Thank you for this interesting suggestion! We have not tried training a single shared model with multiple heads. A priori, it is not clear whether this would increase overall performance, lead to catastrophic interference, or make little difference overall since all three targets are highly related. It is also unclear whether increased model capacity would be needed (or not) and how to combine all three predictions into a single policy (though there are obvious candidates). We believe that investigating these questions is out of scope for our current work, and have added them to a future work section in the discussion of our paper.
>
> Note that we did conduct an ablation where all three predictor targets were trained on the same amount of data in Table A4 to investigate whether our observed differences in performance were simply a matter of different amounts of data.
>
>
> **Why is the state-value predictor not trained on the (much larger) state-action-value dataset, using the state-action values as state-values for the next state?**
>
> Indeed, that is an astute observation, which would cut the annotation time in half when creating the state value and action value datasets (assuming that Stockfish’s evaluation is symmetric). For our paper, we only trained the action value predictor on our largest dataset of ~15 billion data points. However, we also conducted the ablation where all three predictor targets are trained on exactly the same amount of data in Table A4, showing that the state value predictor performs more or less on par (slightly better, by 12 Elo) with the action value predictor. A priori, it seems plausible that learning to predict state and action values is of roughly equal complexity (in deterministic environments), but repeating our experiment from Table A4 (i.e., using the same number of training data points for all policies) at even larger scales would be an interesting direction for future work.

---

> > ### Comment · Reviewer_LZwX · 2024-08-07
> >
> > Thank you for those changes! They clear up all my concerns about the presentation of the motivation/contributions and other details. After more consideration, I've decided to increase my score to 6. In my mind, the main reason against acceptance is still that the paper's main findings aren't very surprising to anyone sufficiently familiar with Lc0, but I think the reasons in favor of acceptance outweigh that:
> > * Most of the ML community is not, in fact, familiar with Lc0, in part because of a lack of scientific/peer-reviewed writings on it. And for this majority of readers, I think the findings are quite interesting and important.
> > * As mentioned, I think this paper is very well done in terms of writing and experiments.
> > * Having a simpler and well-documented/reproducible chess-playing transformer could enable research that would have required significantly more effort on Lc0. So I think this paper has good potential for indirect impact via future work using these models and/or datasets.

---

### Official Review · Reviewer_wU89 · 2024-07-17

**Soundness:** 3
**Presentation:** 3
**Contribution:** 2
**Rating:** 5
**Confidence:** 4

**Summary:**

The paper is well-written and demonstrates that supervised learning with Stockfish evaluations can create a strong chess engine. The methodology is solid, and the inclusion of an LLM as a baseline is noteworthy. However, the motivation is ambiguous and the use of transformers over CNNs is questioned. Additionally, the paper lacks clarity on certain statements, and the technical contribution is limited. Ablation studies on training objectives are also recommended.

**Strengths:**

- This paper is well-written and easy to follow
- It is interesting to know that supervised learning with Stockfish evals can be a strong chess engine baseline
- The methodology is solid and well-explained
- The authors include an LLM as a baseline, it is interesting to see a proper evaluation of how well LLMs can play chess

**Weaknesses:**

- W1: The motivation for this work is somewhat ambiguous. On the one hand, the authors claimed that the main idea is to approximate the search engine Stockfish, by distilling its board evaluation results into NNs (in particular transformers), but the evaluation is on chess performance instead of directly comparing the predicted and ground truth centipawn advantages. Although the action prediction evaluation is kind of doing that, it is still discretized to actions before comparing the rankings, not comparing the continuous values directly. Therefore the claim to approximate stockfish is invalid. On the other hand, the proposed system did really well in chess, it seems that the authors are trying to build a strong chess engine without searching. But it is still underperforming lc0 inference without search.
- W2: Why transformers are particularly interesting for this study? Many prior works have shown that the multi-channel encoding of chess boards combined with CNNs can perform really well, including alphazero, lc0, maia, etc. Essentially, chessboards are 2d and the used inputs are sequential. Is it possible that the tokenization of FEN and the direct inputs to transformers are limiting the performance?
- W3: It is unclear what this statement means: “Although the source code is available, there is no (peer-reviewed) research report explaining and comparing Lc0’s methods”. Does this mean lc0 may include Stockfish eval already? Is it still fair to compare with lc0? It would be good if the authors could clarify.
- W4: The labels of the value prediction are ranges of centipawn advantages predicted by Stockfish. Is there any design to balance the labels? And is there any mechanism to make sure the categorical labels are semantically interconnected?
- W5: The methodological contribution of this work is limited. If I understand correctly, the method basically incorporates stockfish evals as training targets without task-oriented designs.
- W6: Ablation studies on the training objectives should be conducted and presented.

**Questions:**

Please refer to W2-W4.

**Limitations:**

Yes.

---

> ### Author Rebuttal · Authors · 2024-08-06
>
> We thank the reviewer for their constructive feedback.
>
> **The paper claims to aim at approximating Stockfish by distilling its board evaluation into a transformer but evaluates performance w.r.t. chess play rather than Stockfish’s centipawn score, which invalidates this claim. Instead, the paper appears to create a strong chess engine without using explicit search, which, however, underperforms lc0.**
>
> Thank you for pointing out a potential source of confusion!
>
> We do evaluate the learning targets (i.e., the loss over win percentage/centipawns) in Appendix B.4 (Fig. A2 and A3). These loss curves show the quantitative comparison against the discretized ground-truth centipawn score.
>
> We respectfully disagree that our goal was not to distill Stockfish’s value estimation behavior. The objective we *directly* optimize (and report in the appendix) is our model’s prediction error w.r.t. Stockfish’s value estimates, and as a modeling assumption, we chose to phrase this as a discretized prediction problem (similar to distributional RL). This is the main idea behind general algorithm distillation (i.e., minimizing prediction error over an algorithm’s outputs).  We mainly focused on two *behavioral* metrics for comparing models since they imply good quantitative predictions and allow easy comparison across learning targets and models. One could think of these metrics as using a chess-relevant distortion function. If, instead, our goal were to build a strong chess engine, we would have used explicit search and a domain-specific transformer. The fact that lc0’s policy net performs ~160 Elo better is interesting but does not undermine the main goal of our paper (which is to carefully investigate various architectural and training factors, not to outperform).
>
> **Why are transformers interesting for this study if prior work showed that convolutional networks achieve strong performance on chess?**
>
> We focus on investigating transformers’ amortized planning capabilities (in the context of chess). While convolutional networks have been used successfully in the past (e.g., AlphaZero, or past versions of lc0), the more recent consensus seems to be that transformer-based architectures are at least equally suitable, perhaps even stronger. For example, the strongest modern lc0 networks are transformer-based, as also pointed out by  `R-LZwX`, which is the result of many ablations by the lc0 community. While it is intuitive that 2D convolutions implement good inductive biases for the 2D chess board, it is also plausible that (self-)attention leads to good inductive biases for considering relative relations between sets of relevant pieces regardless of their precise spatial arrangement on the board. To avoid speculation, we also performed an ablation with a convolutional network in our paper (see Table A2).
>
> **What does it mean that lc0’s source code is available but that there is no (peer-reviewed) research report explaining and comparing lc0’s methods? Does this mean lc0 may include Stockfish already? Is it still fair to compare with lc0?**
>
> The difficulty with lc0 is that it is a collection of models (originally re-implementing AlphaZero, but with significant evolution since then) that is developed by an active chess engine community, which makes heavy use of the lc0 blog and discord server to share results. This means that detailed architectural information is often only available via the source code (~90k lines of code), and, more importantly, full details about the training process and data sets are typically not part of the open-source release and are thus simply not available for many lc0 networks (though we cannot rule out that this information could be found on lc0’s discord server). For example, even `R-LZwX`, who seems very knowledgeable about lc0, is not fully aware of all the training details. While the lc0 community has made impressive contributions and progress over AlphaZero, the lack of independent reproducibility due to missing information makes the networks currently less suitable for academic research (one of the key requirements of peer-reviewed research articles is providing all information necessary for independent reproduction of results). One positive side-effect that our paper and dataset/model release hopefully have is to encourage and enable others to evaluate their methods in a way that allows for scientific reproducibility.
>
> Consequently, we do not know whether lc0 includes Stockfish evaluations. However, this does not have an impact on our work and main claims. Similarly, whether comparing to lc0 is “fair” cannot be answered without knowing the full training details of lc0, which is a caveat we clearly state in our paper (lines 190 - 192): "We show these comparisons to situate the performance of our models within the wider landscape, but emphasize that some conclusions can only be drawn within our family of models and the corresponding ablations that keep all other factors fixed."
>
> **Can you conduct ablation studies on the training objectives?**
>
> Yes, we have already conducted the following ablations of the training objectives:
> * *loss function* (Table 2): log-loss (classification) vs. HL-Gauss loss (classification) vs. L2 loss (regression)
> * *predictor-target* (Table 2): action-value vs. state-value vs. behavioral cloning
> * *number of value bins* (Table A2): 16 to 256 bins for the HL-Gauss loss
>
> If the reviewer has any other ablation of the training objectives in mind, we are happy to include it in the next revision of our paper.
>
> **Did you try to balance the labels and make sure that they are semantically interconnected?**
>
> We did not balance the labels as we did not want to perturb the natural game/value distribution. We are not quite sure what the reviewer means by “semantic interconnectedness”. Could you please clarify? Note that we did perform ablations on the number of bins we used for our discretization (see Table A2) and an ablation on a non-discretized loss (see Table 2).

---

> > ### Comment · Reviewer_wU89 · 2024-08-13
> >
> > Thanks for the detailed explanations and pointers to the resources in the Appendix.
> >
> > However, I didn't find the answers to "W5: The methodological contribution of this work is limited. If I understand correctly, the method basically incorporates stockfish evals as training targets without task-oriented designs.", which remains my main concern.
> >
> > By "semantic interconnectedness" I mean that: the training targets are stockfish evals and you used "value binning" to cut the value ranges into "classes". However, the "classes" are dependent. For example, the range 0-10 centipawn advantage would be similar to the range 10-20 but significantly to the range 290-300. The method seems without any design to model such dependencies.
> >
> > Nevertheless, the rebuttal has solved most of my concerns. I have adjusted my evaluation.

---

> > > ### Author Response · Authors · 2024-08-14
> > >
> > > We are pleased to hear that the rebuttal has solved most of the reviewer's concerns and thank the reviewer for adjusting their evaluation accordingly!
> > >
> > > We also thank the reviewer for clarifying the term "semantic interconnectedness". We actually spent considerable time addressing the semantic interconnectedness of the classes, and as the reviewer rightly suspected, it turned out to be important for performance. This was precisely the reason for using the HL-Gauss loss [17], which extends the classical cross-entropy loss with label smoothing (see Figure 3 in [18] for an intuitive visualization). We ablated three different loss functions in Table 2: two that model semantic interconnectedness (L2 and HL-Gauss) and one that does not take it into account (i.e., the log loss), with the HL-Gauss loss achieving the highest performance.
> > >
> > > We admit that this important point was not given enough attention in the current manuscript, and we will clarify it in the next revision of our paper.
> > >
> > > We hope that this addresses the reviewer's remaining concerns.

---

### Author Rebuttal · Authors · 2024-08-06

We thank the reviewers for their detailed comments and positive feedback.

We are pleased that the reviewers consider our paper well-written (`R-wU89`, `R-LZwX`, `R-Ye4U`) and a “timely addition to the literature of learning non-trivial algorithms”  (`R-LZwX`, `R-tXJ6`), our methodology solid (`R-wU89`), our experiments extensive “with a wealth of different metrics and interesting ablations” (`R-LZwX`, `R-tXJ6`), and our open-source benchmark dataset “very useful for future research” (`R-Ye4u`, `R-tXJ6`).

Here, we summarize our response to the common questions. We respond to the individual questions below every review.


`R-wU89`, `R-LZwX`, `R-tXJ6`: **What is the motivation for and main contribution of this work?**

The motivation for our work is to understand transformers’ capabilities of amortizing planning via supervised learning. We conduct a case study using chess as a testbed because it is very well studied, memorization is futile even at large scale, and strong chess engines use fairly sophisticated algorithms for which it is not trivially clear that transformers can easily learn to mimic them. As `R-LZwX` points out, the computer chess community has made similar previous and parallel investigations, but not necessarily with the same aim and scientific reproducibility in mind. Hence, we believe that a rigorously executed case study with well-documented details adds value and facilitates future research and scientific comparison of approaches. An important part of this (but not the only contribution) is crafting a benchmark dataset (and documenting exactly how the data is collected and how evaluations are performed).

We have made the following changes (in italics) to our introduction to clarify the motivation and goals of our paper:

* L38: *The goal of our paper is thus to investigate to which degree vanilla transformers can be trained to mimic Stockfish’s value estimation algorithm by minimizing prediction error (log-loss) and what quantitative impact architectural parameters and training design choices have on this capability.* To *scientifically* address this question, we created ChessBench, a large-scale chess dataset created from 10 million human games that we annotated with Stockfish 16.
* L43: The resulting policies are capable of solving challenging chess puzzles and playing chess at a high level (grandmaster) against humans *with Blitz time controls*.
* L51: *We note that building a strong chess engine is not our primary goal (we use game-playing-related metrics, i.e., playing strength and puzzle-solving accuracy, as performance measures since they are highly correlated with good value estimates and allow for easy comparison with other models). Some of our findings are qualitatively known in the chess engine community, particularly the Leela Chess Zero community, and we do compare against one of their state-of-the-art transformer-based models, which performs slightly better than our largest model. Since the training details of this network are opaque, a direct comparison must be taken with a grain of salt, and we instead perform extensive ablations on our model to ensure that all non-ablated parameters are kept fixed. To easily enable further research and improve standardization and reproducibility, we release our dataset and evaluation metrics as a benchmark, ChessBench, and provide some initial results via our models and ablations and the models we compared against.*


`R-wU89`, `R-Ye4U`: **The methodological contributions are limited because there are no domain-specific modifications.**

Indeed, we do not make domain-specific modifications to the architecture/training protocol, but that is deliberate and one of our main design principles: Our goal is to provide an evaluation of transformers in their standard, tried-and-tested setup, rather than building the best possible chess architecture (which would require domain-specific tweaks).

---

### Decision · Program_Chairs · 2024-09-25

**Decision:**

Accept (poster)

**Comment:**

This paper produces a benchmark dataset constisting of chess positions together with Stockfish position evaluations. The main idea is to investigate the ability of transformers to learn this mapping, thus "distilling" Stockfish's search process into a single pass of a transformer. Various dimensions and ablations are investigated, with results showing that a Transformer-only model is capable of playing chess at a high level (without search), although not as well as Stockfish.

The work is solid and was complimented by the reviewers. One strange thing about this paper is that many of the results may indeed be known to the general Leela-chess community, but this would be the first paper that provides documentation and rigour surrounding the idea of using a neural network (in this case, a transformer) to evaluate positions. As such, many of the results will likely be novel to the general machine learning community. On the other hand, one could argue that the audience for this is rather niche. The idea of "distilling" an algorithm or search process into a transformer is generally useful to a wider audience, but because it is so tied to the domain of chess here (naturally, since the benchmark itself focuses on chess), it's unclear whether these results might be of interest outside the domain (and computer chess playing community).

As a minor note, the issue of indeciseveness is tackled in a rather inelegant way, and I can't help but wonder if it reveals an issue with the way the benchmark is constructed. Instead of mapping all guaranteed checkmating sequences to the same value, perhaps they can be mapped to slightly different (high) values to differentiate and encourage the transformer to follow shorter paths to victory.